# An Efficient End-to-End Training Approach for Zero-Shot Human-AI Coordination

**Xue Yan** [1] [2] , **Jiaxian Guo** [3], **Xingzhou Lou** [1] [2], **Jun Wang** [4], **Haifeng Zhang** [1] [2], **Yali Du**[*] [5]

[1]Institute of Automation, Chinese Academy of Science, Beijing, China
[2] School of Artificial Intelligence, University of Chinese Academy of Sciences, China
[3] University of Tokyo, Japan  [4]University College London, UK  [5]King's College London, UK
{yanxue2021,louxingzhou2020,haifeng.zhang}@ia.ac.cn
jiaxian.guo@weblab.t.u-tokyo.ac.jp, jun.wang@cs.ucl.ac.uk, yali.du@kcl.ac.uk

## Abstract

The goal of zero-shot human-AI coordination is to develop an agent capable of collaborating with humans without relying on human data. Prevailing two-stage population-based methods require a diverse population of mutually distinct policies to simulate diverse human behaviors. The necessity of such populations severely limits their computational efficiency. To address this issue, we propose E3T, an **E**fficient **E**nd-to-**E**nd **T**raining approach for zero-shot human-AI coordination. E3T employs a mixture of ego policy and random policy to construct the partner policy, making it both skilled in coordination and diverse. This way, the ego agent is trained end-to-end with this mixture policy, eliminating the need for a pre-trained population, and thus significantly improving training efficiency. In addition, we introduce a partner modeling module designed to predict the partner's actions based on historical contexts. With the predicted partner's action, the ego policy can adapt its strategy and take actions accordingly when collaborating with humans exhibiting different behavior patterns. Empirical results on the Overcooked environment demonstrate that our method substantially improves the training efficiency while preserving comparable or superior performance than the population-based baselines. Demo videos are available at https://sites.google.com/view/e3t-overcooked.

## 1 Introduction

Cooperation between AI agents and humans has gained significant attention, spanning various fields such as robotics [5, 26], automatic driving [41], Human-AI dialogue [14] and Human-AI coordination games [30]. Due to the high cost of collecting human data and involving humans during training, recent studies [32, 15] have focused on zero-shot human-AI coordination problems, with the aim to train an ego agent capable of collaborating with humans without relying on human data.

To train such an ego agent, the prevailing approach [15, 21] involves training the ego agent with diverse partner agents that can simulate human behavior without the need of human data. One method for achieving this is through self-play [33, 29], where the ego agent plays against itself as the partner agent. However, empirical evidence has shown that self-play can lead to agents getting stuck in a single cooperative pattern during training [15], hindering their ability to adapt to diverse human behaviors. Alternative approaches like other-play [15] aim to introduce diversity by breaking the symmetry of self-play policies. Nevertheless, this method relies on the assumption of strict symmetry in the action or state space, limiting its practical applicability.

---

[*]Corresponding author

37th Conference on Neural Information Processing Systems (NeurIPS 2023).

In contrast to relying solely on a single partner policy, advanced population-based methods [21, 42, 32, 18, 19] construct a population with multiple policies to simulate diverse human behaviors, and then train the ego agent to collaborate with this population. Among these methods, FCP [32] obtains a diverse population by training several self-play policies with different random seeds and saving multiple past checkpoints reflecting various coordination levels. TrajeDi [21] promotes diversity by maximizing the divergence of trajectory distributions within the partner population. MEP [42] achieves a cooperative and diverse population by maximizing self-play rewards and population entropy as auxiliary rewards.

However, these population-based methods suffer from two major limitations. Firstly, they are highly computationally inefficient due to their two-stage framework, which involves training a population of policies before training the ego agent. For example, training a MEP agent for one environment in overcooked [8] can take over 15 hours on a single 3090Ti GPU. By contrast, a self-play agent can be trained in less than 1 hour. Secondly, these methods only learn a single response for all partners in the population, neglecting the diverse behavior patterns among them. As a result, the learned ego agent lacks the capability to adjust its policy based on each partner's unique behavior pattern, leading to unsatisfactory performance in zero-shot human-AI coordination.

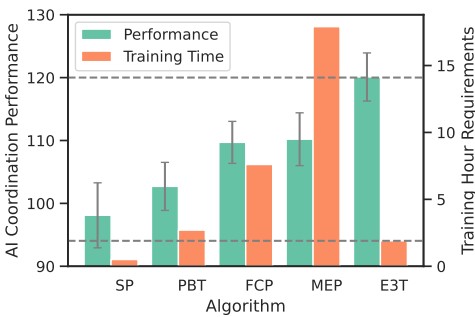

Figure 1: Illustration of training time and zero-shot coordination performance of baselines. The right $y$-axis is the required hours to train one model. The left $y$-axis represents the average rewards of collaborating with AI baselines across 5 layouts.

In this paper, we introduce an **E**fficient **E**nd-to-**E**nd **T**raining approach, called E3T, which aims to improve both efficiency and the ability to recognize behavior patterns in zero-shot human-AI coordination. To achieve efficient training, E3T utilizes a blend of ego and random policies as partners during training, and integrates a partner modeling module into the ego policy. This approach is motivated by the idea that the partner policies should be both skilled in coordination and diverse [42]. By breaking down these two objectives into two policies *i.e.* ego policy and random policy, and merging them as the partner policy, E3T is able to be end-to-end trained without the need for pre-training a population, thus enjoying higher efficiency and a simpler implementation compared to population-based methods.

In order to improve the collaboration capabilities of of ego agents when working with partners exhibiting different behaviors, we introduce a partner modeling module that predicts the partner's actions based on historical contexts. By incorporating this module into the ego policy, the agent is able to know not only the current observation, but also the predicted actions of the partner. This enables the ego agent to adapt its behavior more effectively, aligning it better with the partner's preferences and resulting in improved collaboration performance. To the best of our knowledge, our paper is the first to utilize a partner action prediction module in the context of a two-player human-AI coordination task.

Our contributions are three-fold. Firstly, we introduce an efficient end-to-end ego training approach, named E3T, for zero-shot Human-AI coordination, which offers higher training efficiency compared to population-based methods. Secondly, we propose a partner modeling module that predicts partner actions from the context, allowing the ego policy to adapt when collaborating with humans exhibiting different behavior patterns. Lastly, empirical results on Overcooked show that our method, E3T, significantly improves training efficiency while achieving comparable or superior performance to existing methods as Figure 1 shows. Specifically, E3T demonstrates a significant improvement in training efficiency when compared to state-of-the-art population-based methods.

## 2    Related Work

Our work investigates how to train an ego agent for zero-shot Human-AI coordination, a field with broad applications in real-world scenarios such as robotics [5, 26], service allocation [36, 37, 35] and Human-AI coordination games [30]. In zero-shot Human-AI coordination tasks, reinforcement learning (RL) algorithms [15] are tasked with training a robust ego agent without relying on human data, due to the high cost of collecting real human data.

**Self-Play Training** There are a number of methods to tackle zero-shot coordination, broadly categorized into two classes: self-play and population-based training. Self-play has demonstrated impressive capability in competitive games such as Go [29], Dota [38], StarCraft [34] and imperfect information games [13]. Self-play can also master high-level skills in cooperative games [8]. However, self-play agents often struggle to coordinate effectively with previously unseen partners due to their tendency to adopt rigid conventions formed during training against themselves [32]. Other-play [15] attempts to mitigate these conventions by breaking the symmetries of self-play policies. However, it heavily relies on the prior knowledge of strict symmetries in the environment, limiting its applicability in general environments lacking such symmetries. In this work, E3T follows the self-play training framework, which avoids falling into conventions by encouraging the partner policies to be more diverse. In addition, our method offers greater generality compared to other-play, making it applicable in a wider range of environments.

**Population-Based Training Methods** Population-based methods train a robust AI by collaborating with a diverse population. Experimental results show that population-based methods achieve superior zero-shot coordination performance compared to self-play and other-play [15]. For example, PBT [8] employs an online evolutionary algorithm, which iteratively updates policy parameters and performs policy substitution within the population. FCP [32] achieves diversity within a population by training several self-play policies with different random seeds and saving multiple past checkpoints to obtain agents with various coordination levels. In a similar vein, TrajeDi [21] regulates the partner population to be diverse by maximizing the divergence of trajectory distributions between policies within the population. Furthermore, MEP [42] introduces an auxiliary reward by designing the entropy of the average of policies in the population, thus promoting population diversity. More recently, COLE [18, 19] reformulate two-player coordination as a game-theoretic problem, where the trainer's objective is to estimate the best responses to the most recent population based on the cooperative incompatibility distribution. Our work improves the partner behavior diversity by maximizing the entropy, but E3T directly increases the randomness of a single-agent partner policy, making it more efficient than MEP and TrajeDi.

**Agent Modeling** Agent modeling is an important ability for autonomous agents, enabling them to reason about various properties of other agents such as actions and goals [3]. Typical agent modeling methods focus on reasoning about types of other agents [1, 2, 31], plan recognition [9, 7], group modeling [23, 11, 20] and especially policy reconstruction [6, 25, 4, 27], which is also adopted in E3T to improve zero-shot human-AI coordination performance. However, a key difference is that E3T's ego agent's partner modeling module is trained by pairing with agents and aims to predict the actions of partners. In addition, we directly employ the predicted partner actions for decision-making, instead of using the embedding of partner behaviors [10].

## 3 Preliminary

In this section, we introduce the two-player cooperative Markov Game and a representative pipeline of population-based training for two-player zero-shot coordination.

**Two-player Cooperative Markov Game** A two-player cooperative Markov Game is described by a tuple $\langle \mathcal{S}, \mathcal{A}, \mathcal{A}, \mathcal{T}, R \rangle$, where $\mathcal{S}$ is a finite set of states, $\mathcal{A}$ is the finite set of actions, and suppose two agents have the same action space. $p_0 : \mathcal{S} \to \mathbb{R}$ is the initial state distribution. $\mathcal{T} : \mathcal{S} \times \mathcal{A} \times \mathcal{A} \times \mathcal{S} \to [0, 1]$ is the transition function and assumes the transition dynamic is deterministic. $R : \mathcal{S} \times \mathcal{A} \times \mathcal{A} \to \mathbb{R}$ is the reward function and assume rewards are bounded by $\alpha$, i.e. $R(s, a^1, a^2) \leq \alpha, \forall s, a^1, a^2$. We assume the transition and reward functions are unknown. In this work, we solve the two-policy cooperative Markov Game via single-agent reinforcement learning and the self-play framework as previous studies [8, 15]. Specifically, the ego policy $\pi_e$ is trained by pairing with a fixed partner policy $\pi_p$ with the object defined as $\pi_e = \arg\max_\pi V(\pi, \pi_p)$. The value function of joint policy $(\pi, \pi_p)$ is defined as $V(\pi, \pi_p) = E_{p_0, \mathcal{T}, a^1 \sim \pi, a^2 \sim \pi_p} \sum_{t=0}^{\infty} \gamma^t R(s_t, a_t^1, a_t^2)$, where $\gamma \in (0, 1)$ is the discounted factor.

**Population-Based Training** Population-based methods [21, 42, 32] aim to construct a population of policies to simulate diverse human behaviors. Intuitively, we expect that the policy in the population should be both skilled in coordination and diverse. Taking Maximum Entropy Population (MEP) [42] as an example, it constructs the partner population by maximizing the self-play rewardsand simultaneously increasing the entropy of policies in the population. The objective of Maximum

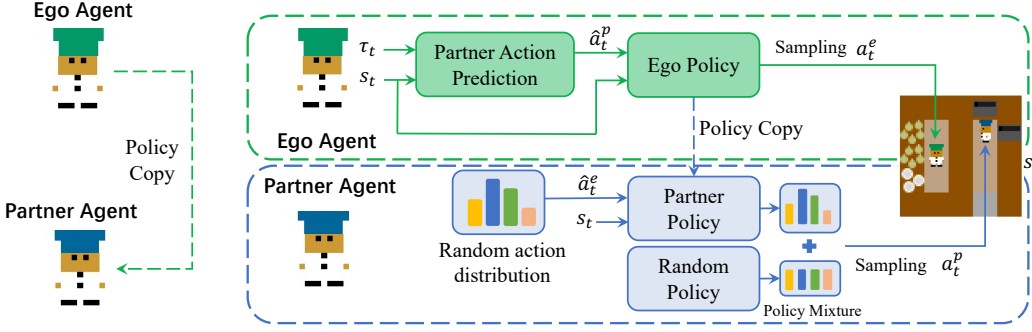

(a) Self-play              (b) E3T

Figure 2: (a). The illustration of the self-play training framework, which trains the ego policy by pairing it with the copied partner policy. (b). The illustration of E3T. The green box shows the decision process of the ego agent. The blue box shows that of the partner agent.

Entropy Population training is defined as:

$$J(\bar{\pi}) = \mathbb{E}_{\pi_i \sim \mathcal{P}} \left[ \sum_t \mathbb{E}_{(s_t, a_t^1, a_t^2) \sim (\pi_i, \pi_i)} \left[ R(s_t, a_t^1, a_t^2) + \lambda \mathcal{H}(\bar{\pi}(.|s_t)) \right] \right]. \tag{1}$$

Here $\mathcal{P} = \{\pi_1, \pi_2, ..., \pi_n\}$ denotes a population of policies, $\bar{\pi}(.|s_t) = \frac{1}{n} \sum_{i=1}^{n} \pi_i(.|s_t)$ is the mean policy of the population $\mathcal{P}$, and $\lambda$ determines the relative importance of the population entropy term against the task reward. The task reward under the joint action $\mathbf{a}_t$ is obtained by self-play with a uniformly sampled policy from the population, and the population entropy term acts as an auxiliary reward. With this competent and diverse population as the partner population, an ego policy is trained by collaborating with the population via prioritized sampling that assigns a higher priority to the partner policy that is difficult to cooperate with. The objective of ego policy training is defined as:

$$J(\pi_e) = \sum_t \mathbb{E}_{(s_t, a_t^1, a_t^2) \sim (\pi_e, \rho(\mathcal{P}))}[R(s_t, a_t^1, a_t^2)], \tag{2}$$

where $\rho(.)$ as the prioritized sampling mechanism. However, the training process of MEP is inefficient due to the complex two-stage training and the necessity of maintaining a large population for behavioral diversity. Our work draws inspiration from MEP, which achieves behaviorally diverse and cooperative partners by maximizing the self-play reward and the entropy of partner policy and then trains a robust ego policy by cooperating with those partners. In the next section, we will discuss how to solve zero-shot coordination using an efficient end-to-end training process.

## 4 Methodology

In this section, we introduce the proposed method, E3T, an efficient end-to-end training approach, which trains an ego policy $\pi_e$ through collaborating with a partner policy $\pi_p$. To take into account both training efficiency and partner diversity, we employ a mixture of the copied ego policy and a random policy as the partner policy $\pi_p$. The objective of E3T is defined as follows:

$$J(\pi_e) = \sum_t \mathbb{E}_{(s_t, a_t^1, a_t^2) \sim (\pi_e, \pi_p)}[R(s_t, a_t^1, a_t^2)] \tag{3}$$

The proposed method contains two key components, a diverse partner policy and a partner modeling module for adapting to unseen partners, which will be introduced in Section 4.1 and Section 4.2 respectively. An overview of E3T is presented in Figure 2.

### 4.1 A Mixture Policy as Partner

We begin with the motivation of leveraging the proposed mixture partner policy. Recall that MEP trains a partner population by maximizing self-play rewards of policies within it and maximizing the population entropy as shown in Equation (1). The first term on maximizing self-play rewards tends to learn a population with cooperative ability. The second entropy term can force the partner population

to be diverse. However, optimizing the objective of Equation (1) suffers very high computational complexity due to the necessity of training a number of policies to collaborate effectively with themselves. Moreover, it exhibits unstable convergence properties, requiring meticulous hyperparameter tuning for additional entropy term [12].

To circumvent computational cost and non-stable training, we construct the partner policy via dividing Equation (1) into two parts with different partner policies learned, *i.e.* $\pi_{p1}$ and $\pi_{p2}$, which aim to maximize the rewards $\sum_t \mathbb{E}_{(s_t, \mathbf{a}_t) \sim \pi_{p1}, \pi_{p1}} [R(s_t, \mathbf{a}_t)]$ and the entropy of the policy output $\sum_t \mathbb{E}_{(s_t, \mathbf{a}_t) \sim \pi_{p2}} [\mathcal{H}(\pi_{p2}(.|s_t))]$, respectively. We can see that maximizing the former objective function is aligned with the goal of learning coordination skills, akin to the self-play training objective. In this way, we can directly use the ego policy $\pi_e$ as $\pi_{p1}$. On the other hand, if we directly learn the latter one objective, *i.e.*, the entropy of the mean policy, policy $\pi_{p2}$ will lead to the random policy $\pi_r$, *i.e.*, a uniform distribution over the action space.

To construct a partner who has both skills in coordination and diverse behaviors, we directly mix the above-discussed partner policies $\pi_{p1}$ and $\pi_{p2}$ as the partner policy $\pi_p$. As previously discussed, $\pi_{p1}$ can directly apply ego policy $\pi_e$ to achieve high reward for coordination. $\pi_{p2}$ is a random policy $\pi_r$, which has the maximum policy entropy. In this way, the partner policy $\pi_p$ is the mixture between the self-play ego policy and the random policy with the balance parameter $\epsilon$, formally defined as:

$$\pi_p = \epsilon \pi_r + (1 - \epsilon) \pi_e \tag{4}$$

We can see that the computation cost for obtaining this mixture partner policy is significantly lower than population-based methods since the conversion eliminates the need to train a population of policies. Additionally, the partner policy achieves behavioral diversity by simply leveraging a random policy with maximum entropy, whereas population-based approaches rely on more complex mechanisms, such as gradient descent [21] or reward shaping [42], to enforce the diversity. Both diversity and coordination ability are two keys to constructing a partner policy. Intuitively, the diversity reflected by entropy increases, and the coordination performance decreases with the random coefficient epsilon increasing. We attempt to quantitatively verify these intuitions through Propositions 1 and 2. Firstly, we analyze the behavior diversity of the partner policy of E3T. The entropy of partner policy reflects the diversity of partner behaviors. Proposition 1 shows that the entropy of the mixture partner policy is related to that of the random policy with maximum entropy.

**Proposition 1** (Entropy of Mixture Partner Policy). *The entropy of the mixture partner policy, defined as $\pi_p = (1 - \epsilon)\pi_e + \epsilon\pi_r$, satisfies that:*

$$\mathcal{H}(\pi_p(.|s_t)) \geq \epsilon \mathcal{H}(\pi_r(.)) - C_1, \forall s_t \tag{5}$$

*where $C_1 = \epsilon \log \epsilon + |\mathcal{A}|(1 - \epsilon)$, and $|\mathcal{A}|$ is the cardinality of the action apace $\mathcal{A}$. In addition, the lower bound of $\mathcal{H}(\pi_p(.|s_t))$ is increased as the increasing of $\epsilon$. Proof. See Appendix.*

Secondly, we demonstrate the benefits of improving coordination skills when pairing with the partner policy of E3T. According to Proposition 2, although $\pi_e'$ is the best response to the mixture partner policy rather than the ego policy itself, the coordination performance of $\pi_e'$ with the ego policy is lower bounded by that of ego policy self-play with a punishment term $\zeta$. The $\zeta$ is related to the balance parameter $\epsilon$ and smaller $\epsilon$, which means less weight on the random policy, leads to lower $\zeta$.

**Proposition 2** (Coordination Skill Improvement). *Let the $\pi_e'$ as the best response to $\pi_p = (1 - \epsilon)\pi_e + \epsilon\pi_r$, i.e $\pi_e' = \arg\max_\pi V(\pi, \pi_p)$, then we have $V(\pi_e', \pi_e) \geq V(\pi_e, \pi_e) - \zeta$, where $\zeta = \frac{4\alpha\epsilon}{(1-\gamma)^2}$. Proof. See Appendix.*

## 4.2 Partner Modeling

In this section, we describe the details of the proposed partner modeling module, designed to extract and analyze the behavior pattern of the partner from historical information. This module equips the ego policy with the ability to consider not only the current state but also the partner's behavior pattern, empowering the ego agent to adapt to previously unseen partners

**Partner Action Prediction** As in previous works [6, 25, 4], we model the partner agent by policy reconstruction. If the actual partner action distribution $\mathbf{a}_t^p$ is given, the two-player cooperative Markov Game will reduce to a single-agent MDP about the ego agent, in which the ego agent can easily find the best response to the current state. Formally, under the partner policy $\pi_p$ with $\mathbf{a}_t^p = \pi_p(.|s_t)$, the two-player MDP with transition function $\mathcal{T}$ can be converted in to a single-agent MDP with transition

**Algorithm 1** E3T: An efficient end-to-end training approach with mixture policies and partner modeling

---

**Input:** Parameters of context encoder $E_c$, action prediction network $F$, ego policy $\pi_e$, hyper-parameter $\epsilon$, $k$.
**Output:** Well-trained context encoder $E_c$, action prediction network $F$ and the ego policy $\pi_e$.
 1: **while** not done **do**
 2:     Set data buffer $D = \emptyset$.
 3:     **for** $t = 1, 2, \cdots$, step **do**
 4:         Encode context of the partner: $z_t^c = E_c(\tau_t)$
 5:         Predict partner action distribution: $\hat{\mathbf{a}}_t^p = F(z_t^c)$
 6:         Randomly select an action distribution: $\hat{\mathbf{a}}_t^e$
 7:         Mixture partner policy as: $\pi_p(\cdot|s_t, \hat{\mathbf{a}}_t^e) = (1 - \epsilon)\pi_e(\cdot|s_t, \hat{\mathbf{a}}_t^e) + \epsilon\pi_r(\cdot)$
 8:         Gather data from $\pi_e(\cdot|s_t, \hat{\mathbf{a}}_t^p)$ and $\pi_p(\cdot|s_t, \hat{\mathbf{a}}_t^e)$, then $D = D \cup \{(\tau_{t+1}, s_t, a_t^e, a_t^p, r_t)\}$
 9:     **end for**
10:     Update parameters of partner modeling module by minimizing Equation (7).
11:     Update parameters of the ego policy $\pi_e$ via PPO.
12: **end while**

---

function denoted as $\mathcal{T}_{\pi_p}(s'|s_t, a) = \sum_{a^p} \pi_p(a^p|s_t)\mathcal{T}(s'|s_t, a^p, a)$. However, because the agents act simultaneously, the ego agent can only infer the partner's action from historical information instead of directly observing the actual partner's action.

To obtain the predicted partner action $\hat{\mathbf{a}}_t^p$, we first devise an encoder network $E_c$ to extract historical partner context $z_t^c$ from a sequence of past state-action pairs $\tau_t = \{(s_{t-i}, a_{t-i}^p)_{i=1}^k\}$. The historical partner context $z_t^c$ contains rich information of the partner's behavior and coordination pattern, enabling our action prediction network $F$ to predict the partner's action distribution based on $z_t^c$ and current state $s_t$. This procedure is formally written as follows:

$$z_t^c = E_c(\tau_t), \ \hat{\mathbf{a}}_t^p = F(s_t, z_t^c) \tag{6}$$

The parameters of the context encoder $E_c$ and the action prediction network $F$ are jointly optimized by minimizing a cross-entropy loss:

$$\mathcal{L} = \sum_{i=1}^{N} -\mathbf{y}_i^T \log \hat{\mathbf{a}}_i^p, \tag{7}$$

where $N$ is the batch size, and $\mathbf{y}_i$ is a one-hot vector indicating the actual partner action.

We provide a theoretical analysis of how the accuracy of partner modeling affects the coordination ability as shown in Proposition 3. We analyze the coordination performance gap between the best responses to the ground-truth partner policy and the partner prediction module. It is established that this performance gap is proved to be proportional to the prediction error.

**Proposition 3.** *Denote $\pi_\tau(.|s_t) = \mathbb{E}_{\tau_t}[F(s_t, E_c(\tau_t))]$ as the expectation of the proposed partner prediction module over historical trajectory $\tau_t$. The ground-truth partner policy is denoted as $\pi_p(.|s_t)$. Assume the learned $\pi_\tau(.|s_t)$ satisfies that $D_{TV}(\pi_\tau(.|s_t), \pi_p(.|s_t)) \leq \epsilon_m, \forall s_t$. Define the $\mu_\tau^*$ and $\mu_p^*$ are the best response to the partner prediction module and the ground-truth partner policy respectively. Then the value functions of these best responses collaborating to the partner policy satisfy that: $|V(\mu_\tau^*, \pi_p) - V(\mu_p^*, \pi_p)| \leq \frac{2\gamma\epsilon_m\alpha}{(1-\gamma)^2}$. Proof. See Appendix.*

### 4.3 Ego Policy Training and Test

E3T separately trains the partner action prediction module via minimizing the loss function in Equation (7) and trains the ego policy that makes decisions depending on both observations and predicted partner action distribution via PPO [24]. The detailed training process of E3T is described in the Algorithm 1. During testing time, the ego agent zero-shot coordinates with an unseen partner, without parameters updating or temporally-extended exploration as [22].

**Implement Details** The partner modeling module of E3T takes current observation at step $t$ and partner's past $k$ state-action pairs as input as shown in Equation (6), we set $k = 5$ in this work. The ego policy $\pi_e(.|s_t, \hat{\mathbf{a}}_t^p)$ conditions on current state $s_t$ and predicted partner action distribution $\hat{\mathbf{a}}_t^p$,

Table 1: The cardinality of the state space of 5 layouts. The state space's cardinality of *Counter Circuit* is up to $5.8 \times 10^{16}$.

| Layout | Cramped Rm. | Asymm. Adv. | Coord. Ring | Forced Coord. | Counter Circ. |
|---|---|---|---|---|---|
| State Space | $5.3 \times 10^7$ | $1.2 \times 10^{14}$ | $6.8 \times 10^{10}$ | $2.0 \times 10^9$ | $5.8 \times 10^{16}$ |

Table 2: Comparisons of baselines in terms of training time and coordination performance with behaviour-cloned human proxies. The training time represents the hours required for training one model on the layout *Cramped Rm* (training time on other layouts is similar). We report the average and the standard error of coordination rewards over 5 random seeds.

| Baseline | SP | PBT | FCP | MEP | E3T |
|---|---|---|---|---|---|
| Training Time(h) | 0.5 | 2.7 | 7.6 | 17.9 | 1.9 |
| Average Reward | $65.8 \pm 5.5$ | $66.8 \pm 5.0$ | $71.7 \pm 3.7$ | $89.7 \pm 7.1$ | $\mathbf{114.1 \pm 6.4}$ |

which enables the ego agent to analyze and adapt to its partner, while a random action distribution $\hat{a}_t^e$ is fed into the partner policy as the predicted ego agent policy. The random action distribution breaks symmetries and conventions commonly seen in self-play agents and introduces randomness into the partner policy, which in turn improves partner diversity. Moreover, this is consistent with the fact that some humans may not take their AI partners' actions into consideration, but the AI agents should adapt to their human partners.

# 5 Experiments

In this section, we evaluate E3T on zero-shot collaborating with behavior-cloned human proxies, real humans, and AI baselines. More details of the experiment setting and additional results can be found in Appendix.

## 5.1 Environments

We evaluate the zero-shot coordination performance of E3T on a $100 \times 100$ cooperative matrix game [21, 42] and the Overcooked environment [8]. Overcooked has a discrete action space with 6 actions, and we conduct experiments on 5 different layouts that pose various coordination challenges, such as avoiding blocking each other and preparing the required ingredients for a soup together. In addition, the state spaces of 5 layouts are different as shown in Table 1. For instance, the state space of *Counter Circuit* reaches up to $5.8 \times 10^{16}$. This statistic indicates the 5 layouts are challenging for explorations. We have

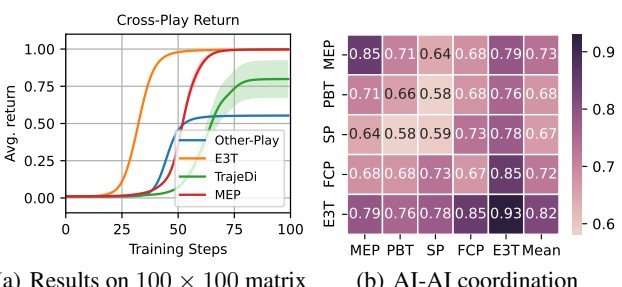

(a) Results on $100 \times 100$ matrix game

(b) AI-AI coordination

Figure 3: (a) We report the cross-play returns between 10 independently trained models for each baseline. (b) These results show the normalized coordination rewards between baselines averaged over 5 layouts.

conducted additional experiments on the Google Football [17] environment's "3 vs 1 with Keeper" layout, which is a three-player cooperative game with a discrete action space comprising 19 actions. More details about the environment descriptions can be found in Appendix.

## 5.2 Baselines

We compare proposed E3T with 6 baselines: Self-Play (SP) [28], Population-Based Training (PBT) [16, 8], Fictious Co-Play (FCP) [32], Trajectory Diversity (TrajeDi) [21], Maximum-Entropy Population-based training (MEP) [42] and Other-Play (OP) [15]. All methods, except SP, OP, and E3T, fall under the category of population-based methods, which aim to develop a robust ego policy through interaction with a partner population.Detailed descriptions of baselines and the network details of E3T can be found in Appendix.

## 5.3   Result Analysis

We compare E3T with other baselines on zero-shot collaboration with the human proxy, real humans, and AI baselines.

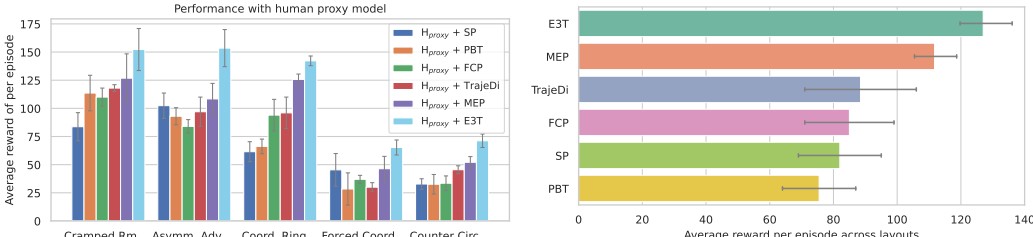

(a) Results on coordinating with human proxy models      (b) Results on collaborating with real humans

Figure 4:   (a) We plot the mean and standard error of coordination rewards over $5$ random seeds. The parameter $\epsilon$ is set to $0.5$ for all layouts except *Forced Coord.*, where $\epsilon$ is settoas $0.0$. In this layout, the ego agent does not need too much exploration due to the narrow active range. (b) We plot the mean and standard error of humans collaborating with baselines over all layouts. These results show that E3T outperforms all other baselines when collaborating with real humans.

**Zero-shot Coordination on** $100 \times 100$ **matrix game**   Figure 3(a) presents the cross-play[15] results of $10$ independently trained policies for each baseline on a one-step matrix game. The cross-play result measures the coordination performance between models that belong to the same baseline but have different random seeds. In this game, two players select a row and a column independently, and the reward is determined by the intersection of their choices. As shown in Figure 3(a), E3T and MEP can converge to the optimal strategies with the largest return as $1$, while E3T can converge faster than MEP. These results indicate that the mixture partner policy can enable sufficient exploration and avoid getting stuck in local optima that generalize poorly on the game with large action space. More details of the matrix game are given in Appendix.

**Training Efficiency Analysis**   Table 2 compares the training time and average rewards of different baselines, where the rewards are calculated by averaging over $5$ layouts on collaboration with human proxies. These results suggest that E3T outperforms existing baselines on collaboration with human proxies, and E3T is more efficient than population-based methods. For instance, it only spends $10.6\%$ training time of previous state-of-the-art MEP.

**AI-AI Coordination**   The zero-shot coordination performance between AI baselines are shown in Figure 3(b), where each element represents the normalized cooperation rewards averaged over $5$ layouts for a pair of baselines. The performance of baselines cooperating with themselves is measured by the cross-play metric. As shown in Figure 3(b), E3T outperforms other baselines in coordinating with AI models. In addition, SP and PBT, which train policies only based on rewards and ignore diversity, have poor generalization to unseen AI partners. MEP and FCP, which train the ego agent by best responding to a pre-trained diverse partner population, perform better than SP and PBT, but worse than E3T. These results indicate that partner diversity is essential for zero-shot coordination, and E3T learns a roust ego policy by providing sufficient partner diversity and being equipped with adaptation ability. More results of AI-AI coordination on $5$ layouts can be found in Appendix.

**Human Proxy-AI Coordination**   Figure 4(a) shows the results of baselines cooperating with behavior-cloned human proxies. These results suggest that E3T consistently outperforms existing baselines on all layouts. In addition, FCP, MEP, and TrajeDi have gained better or similar generalization performance than SP and PBT, owing to training an ego agent based on a diverse partner policy. These results indicate that the diversity of partners is important for the zero-shot coordination task. Note that SP achieves comparable performance to population-based MEP on *Forced Coord.* and E3T achieves the best coordination performance at parameter $\epsilon = 0.0$. One possible reason is that the *Forced Coord.* layout is a difficult layout with a very narrow activity range and two agents being forced to coordinate to complete a soup. Therefore, it requires more cooperation than exploration. Moreover, E3T with $\epsilon = 0.0$ is better than SP, indicating that the partner modeling module is beneficial to adapting to an unseen partner.

**Human-AI Coordination**   The average rewards over $5$ layouts of AI baselines coordinating with real humans are shown in Figure 4(b). We follow the Human-AI coordination testing protocol

proposed in [32] and recruit 20 individuals to evaluate the Human-AI coordination ability of E3T and state-of-the-art MEP, where individuals play a total of 600 rounds with AI. For cost and time consideration, we reuse the Human-AI coordination results of other baselines reported in [8, 42]. Figure 4(b) shows that E3T outperforms all other baselines when zero-shot coordinating with real humans. We also collect individuals' subjective assessment of E3T and MEP about the degree of intelligence, collaborative ability, and preference between the two. More details of the Human-AI coordination setting and individuals' subjective assessment are given in Appendix.

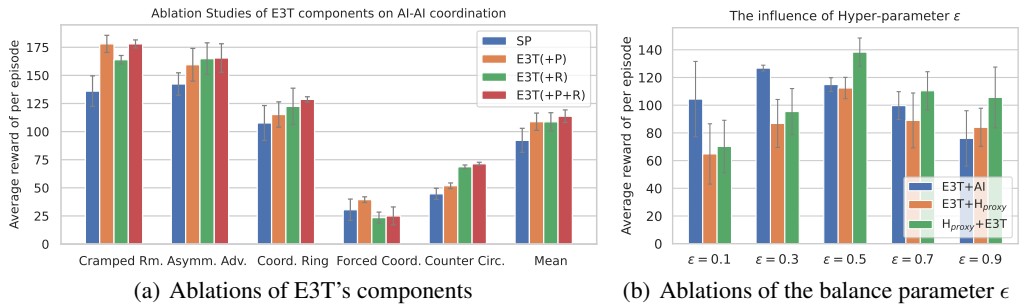

| (a) Ablations of E3T's components | (b) Ablations of the balance parameter $\epsilon$ |

Figure 5: (a) Ablation of E3T's components on collaborating with AI baselines. E3T(+R) and E3T(+P) respectively integrate the mixture partner and the partner modeling module into the self-play framework. The balance parameter $\epsilon$ is set to $0.3$ for E3T(+R). (b) Results on E3T with different balance parameters $\epsilon$. We plot the average rewards over 5 random seeds when E3T collaborating with AI (MEP, PBT) baselines and human proxy on layout *Coord. Ring.* The blue bar shows performance of E3T cooperating with AI baselines, while the orange and the green bars show its performance when cooperating with the human proxy, with starting positions switched.

## 5.4 Ablation Study

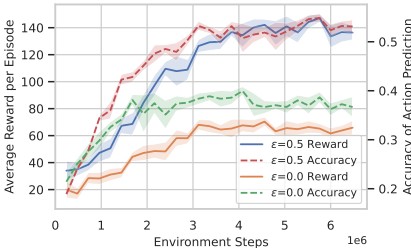

Figure 6: Relationship between the accuracy of partner action prediction and coordination performance when collaborating with the human proxy on layout *Coord. Ring.* We plot the rewards and prediction accuracy obtained by coordinating with the human proxy at different checkpoints along the training process.

**Analysis of Component Ablation Results** We present the ablation results of E3T's components in Figure 5(a). We evaluate the effectiveness of each component on collaborating with AI baselines (including MEP, FCP, and PBT). We use (+R) and (+P) to denote models incorporated with the mixture partner policy and the partner modeling module, respectively. As shown in Figure 5(a), E3T(+P) achieves a higher average reward than SP on each layout, which shows that the partner modeling module can improve zero-shot coordination by apdating actions accordingly. Likewise, E3T(+R) surpasses SP on 4 layouts except for the *Forced Coord.*, which shows that encouraging more exploration in the partner policy can discover more coordination patterns during training and thus can enhance zero-shot coordination with unseen partners during testing. The lower performance on *Forced Coord.* may be due to its narrow action space and the lack of reward incentives for coordination when using a too random partner policy during training. Moreover, E3T(R+P) that combines both the mixture partner policy and the partner modeling module performs better than E3T(R) and E3T(P) on the average reward over all layouts.

**Analyzing the effect of the hyper-parameter** $\epsilon$ Figure 5(b) shows how the balance parameter $\epsilon$ of the mixture partner affects the performance of E3T when collaborating with AI baselines and the human proxy. We observe that the coordination rewards of E3T increase and then decrease as $\epsilon$ increases for both types of partners. The results indicate that both collaboration ability and partner diversity are essential for zero-shot coordination, but simply increasing the partner action diversity is not enough. The optimal values of $\epsilon$ are 0.3 and 0.5 for AI baselines and human proxy, respectively. One possible reason for this discrepancy is that the human proxy tends to generate more 'noop' actions [8], which reduces the need for coordination behaviors compared to AI baselines.

Table 3: We present the coordination performance of baselines during training and zero-shot testing on a three-player Google Football game, where the coordination performance is measured by the winning rates of scoring a goal. We independently train 3 policies with different random seeds for each baseline. For zero-shot testing, we consider all 6 possible permutations of the 3 trained policies and report the mean and standard error of the winning rates as Zero-shot (Mean). We also report the highest winning rate among the permutations as Zero-shot (Max).

|  | Training Performance | Zero-shot (Mean) | Zero-shot (Max) |
|---|---|---|---|
| Self-play | $0.79 \pm 0.08$ | $0.07 \pm 0.08$ | 0.24 |
| E3T(mixture policy) | $0.76 \pm 0.09$ | $0.49 \pm 0.21$ | 0.78 |

**Analyzing the effect of partner modeling module** To demonstrate the effect of the proposed partner modeling module, we plot the accuracy of predicting partner actions and the rewards when cooperating with a human proxy. As shown in Figure 6, the reward and the prediction accuracy show the same tendency: better partner modeling leads to improvement in coordination. In addition, the accuracy at $\epsilon = 0.5$ reaches around $55\%$, which is higher than the accuracy around $40\%$ when $\epsilon = 0.0$. This indicates that mixing a random policy can provide more exploration and thus generalize well to unseen partners.

**Does the mixture partner policy accommodate the multi-player game?** We verify the effect of the mixture partner policy on improving the zero-shot coordination performance on a three-player Google Football game, the 3 vs 1 with Keeper scenario. We train self-play policies via the Multi-Agent PPO (MAPPO) [40], where policies have shared parameters. E3T(mixture policy) integrates the mixture partner policy into the self-play training process, in detail, the three agents randomly sample an action with a probability as $\epsilon$, otherwise, take actions according to the action policies. We independently train 3 policies with different random seeds for each baselines and report the cross-play results between unseen policies to test the zero-shot coordination ability of these models.

Table 3 shows that self-play policies fail to zero-shot cooperate with policies from different training seeds, because they may fall into some specific coordination conventions during training. Our proposed mixture partner policy can increase the behavioral diversity of the partners, allowing the ego policy to encounter different coordination patterns during training. Consequently, our method with the mixture partner policy can train robust ego policies that can adapt well to policies independently trained from different training seeds. Moreover, increasing partner's behavior diversity during training can significantly improve the generalization ability of the trained ego policy.

# 6 Conclusion

In this paper, we propose an **E**fficient **E**nd-to-**E**nd **T**raining approach for zero-shot human-AI coordination, called E3T. Different from previous population-based methods, E3T employs a mixture policy of ego policy and random policy to construct the partner policy, enabling the partner policy to be both coordination-skilled and diverse. In this way, the ego-agent is able to be end-to-end trained in such a mixture policy without pre-training a population, thus significantly improving the training efficiency. Additionally, we introduce a partner modeling module that allows the ego agent to predict the partner's next action, enabling adaptive collaboration with partners exhibiting different behavior patterns. Empirical results on Overcooked clearly demonstrate that E3T can significantly improve training efficiency while achieving comparable or superior performance compared to existing methods.

**Limitations and Future Work** In this paper, we only explore the feasibility of our framework in zero-shot two-player human-AI coordination tasks but neglect general multi-player (more than two) human-AI coordination tasks. Future research will focus on expanding the framework to multi-player (more than two) human-AI coordination tasks, with a goal of enhancing both zero-shot coordination performance and training efficiency. Since AI agents are unable to regulate human behavior, indiscriminately collaborating with humans in accomplishing their goals could result in unlawful circumstances. Therefore, future research should concentrate on developing AI agents capable of detecting dangerous intentions and preventing their misuse for illicit or military purposes.

## Acknowledgements

We would like to thank Yang Li for providing the models for the MEP and FCP baselines. Haifeng Zhang thanks to the support of the NSFC Grant Number 62206289.

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

# A Proposition Proofs

## A.1 Proof of Proposition 1

**Proposition 1** (Entropy of Mixture Partner Policy). *The entropy of the mixture partner policy, defined as $\pi_p = (1 - \epsilon)\pi_e + \epsilon\pi_r$, satisfies that:*

$$\mathcal{H}(\pi_p(.|s_t)) \geq \epsilon\mathcal{H}(\pi_r(.)) - C_1, \forall s_t \tag{5}$$

*where $C_1 = \epsilon \log \epsilon + |\mathcal{A}|(1 - \epsilon)$, and $|\mathcal{A}|$ is the cardinality of the action apace $\mathcal{A}$. In addition, the lower bound of $\mathcal{H}(\pi_p(.|s_t))$ is increased as the increasing of $\epsilon$. Proof. See Appendix.*

*Proof.* The mixture partner policy is defined as $\pi_p = (1 - \epsilon)\pi_e + \epsilon\pi_r$ with the balance parameter $\epsilon$. Detailed derivations of Equation (5) in Proposition 1 are as follows:

$$
\begin{aligned}
&\mathcal{H}(\pi_p(.|s_t)) \\
&= \sum_a -\pi_p(a|s_t) \log \pi_p(a|s_t) \\
&= \sum_a -\left[\epsilon\pi_r(a) + (1 - \epsilon)\pi_e(a|s_t)\right] \log \pi_p(a|s_t) \\
&= \sum_a -\epsilon\pi_r(a) \log \pi_p(a|s_t) - (1 - \epsilon)\pi_e(a|s_t) \log \pi_p(a|s_t) \\
&\overset{(1)}{\geq} \sum_a -\epsilon\pi_r(a) \log \pi_p(a|s_t) \\
&= \sum_a -\epsilon\pi_r(a) \log \left[\epsilon\pi_r(a)\left(1 + \frac{(1 - \epsilon)\pi_e(a|s_t)}{\epsilon\pi_r(a)}\right)\right] \\
&\overset{(2)}{\geq} \sum_a -\epsilon\pi_r(a) \left[\log \epsilon + \log \pi_r(a) + \frac{(1 - \epsilon)\pi_e(a|s_t)}{\epsilon\pi_r(a)}\right] \\
&= \sum_a -\epsilon\pi_r(a) \log \epsilon - \epsilon\pi_r(a) \log \pi_r(a) - (1 - \epsilon)\pi_e(a|s_t) \\
&= \epsilon\mathcal{H}(\pi_r(.)) - \epsilon \log \epsilon - |A|(1 - \epsilon)\pi_e(a|s_t) \\
&\overset{(3)}{\geq} \epsilon\mathcal{H}(\pi_r(.)) - \epsilon \log \epsilon - |A|(1 - \epsilon) \\
&= \epsilon\mathcal{H}(\pi_r(.)) - C_1,
\end{aligned} \tag{8}
$$

where $C_1 = \epsilon \log \epsilon + |\mathcal{A}|(1 - \epsilon)$, and $|\mathcal{A}|$ is the cardinality of the action apace $\mathcal{A}$. Inequality (1) and (3) hold because $\pi_p(a|s_t) \leq 1$ and $\pi_e(a|s_t) \leq 1$. Inequality (2) holds because $log(1 + x) \leq x$. Since $C_1$ decreases in the range $\epsilon \in (0, e^{|A|-1}]$ and $\epsilon$ belongs to $[0, 1]$, the lower bound of $\mathcal{H}(\pi_p(.|s_t))$ increases as $\epsilon$ increases.

$\square$

## A.2 Necessary Notations and Lemmas for Proofs of Proposition 2 and 3

**Notations** As the previous statement in section 4.2, that when the partner policy $\pi_p$ is given with $\mathbf{a}_t^p = \pi_p(.|s_t)$, the two-player MDP with transition function $\mathcal{T}$ can be converted in to a single-agent MDP with transition function denoted as $\mathcal{T}_{\pi_p}(s'|s_t, a) = \sum_{a^p} \pi_p(a^p|s_t)\mathcal{T}(s'|s_t, a^p, a)$. Note that the value estimation $V(\pi_1, \pi_2)$ under the two-player transition function $\mathcal{T}$ is equal to the value estimation $V^{\mathcal{T}_{\pi_2}}(\pi_1) = \frac{1}{1-\gamma}\mathbb{E}_{(s,a^1)\sim\rho_{\pi_1}(\mathcal{T}_{\pi_2})}[\sum_{a^2}\pi_2(a^2|s)R(s, a^1, a^2)]$ under the single-agent transition function $\mathcal{T}_{\pi_2}$. $\rho_{\pi_1}(\mathcal{T}_{\pi_2})$ denotes the stationary state-action distribution under the single-agent transition function $\mathcal{T}_{\pi_2}$, formally written as:

$$\rho_{\pi_1}(s, a^1; \mathcal{T}_{\pi_2}) = (1 - \gamma) \sum_{t=0}^{\infty} \gamma^t \mathbb{P}(s_t = s, a_t^1 = a^1; \pi_1, \mathcal{T}_{\pi_2}). \tag{9}$$

Then, the equivalence between the the value estimations $V(\pi_1, \pi_2), V^{\mathcal{T}_{\pi_2}}(\pi_1)$ is proven as:

$$
\begin{aligned}
V^{\mathcal{T}_{\pi_2}}(\pi_1) &= \frac{1}{1-\gamma} \mathbb{E}_{(s,a^1)\sim\rho_{\pi_1}(\mathcal{T}_{\pi_2})} \Big[\sum_{a^2} \pi_2(a^2|s) R(s, a^1, a^2)\Big] \\
&= \sum_{s,a^1,a^2} \sum_{t=0}^{\infty} \gamma^t \mathbb{P}(s_t = s, a_t^1 = a^1; \pi_1, \mathcal{T}_{\pi_2}) \pi_2(a^2|s) R(s, a^1, a^2) \\
&= \sum_{s,a^1,a^2} \sum_{t=0}^{\infty} \gamma^t \mathbb{P}(s_t = s, a_t^1 = a^1, a_t^2 = a^2; \pi_1, \pi_2, \mathcal{T}) R(s, a^1, a^2) \\
&= V(\pi_1, \pi_2)
\end{aligned}
\tag{10}
$$

We next give the necessary lemmas for proofs of Proposition 2 and 3. We first introduce the following Lemma, which describes the relationship between value estimation and policy distribution discrepancy on single-agent MDP.

**Lemma 1.** *Given a single agent MDP with transition $\mathcal{T}$, assume there are two policies $\pi_a$ and $\pi_b$ with $D_{TV}(\pi_a(.|s), \pi_b(.|s)) \leq \epsilon, \forall s, a$ holding. Then the difference between value functions satisfies that $|V^{\mathcal{T}}(\pi_a) - V^{\mathcal{T}}(\pi_b)| \leq \frac{2\alpha\epsilon}{(1-\gamma)^2}$, where rewards are bounded by $\alpha$ and $\gamma$ is the discounted factor.*

*Proof.* The Lemma 1 can be proved directly following the proof process of Theorem 1 in [39]. □

We provide the error bound on value estimations under different transitions used for the proof of Proposition 3.

**Lemma 2.** *Given an MDP with true transition function $\mathcal{T}^*$ and assume the estimated transition function $\mathcal{T}'$ satisfies that, for a policy $\pi$, $\mathbb{E}_{\rho_{\pi(\mathcal{T}^*)}}[D_{TV}(\mathcal{T}^*(.|s,a), \mathcal{T}'(.|s,a))] \leq \epsilon_m$, then we have:*

$$
\left| V^{\mathcal{T}'}(\pi) - V^{\mathcal{T}^*}(\pi) \right| \leq \frac{\alpha\gamma}{(1-\gamma)^2} \epsilon_m,
\tag{11}
$$

*where rewards are bounded by $\alpha$ and $\gamma$ is the discounted factor*

*Proof.* The Lemma 2 can be proved directly following the prove process of Lemma 7 in [39]. □

### A.3 Proof of Proposition 2

**Proposition 2** (Coordination Skill Improvement). *Let the $\pi'_e$ as the best response to $\pi_p = (1-\epsilon)\pi_e + \epsilon\pi_r$, i.e $\pi'_e = \arg\max_\pi V(\pi, \pi_p)$, then we have $V(\pi'_e, \pi_e) \geq V(\pi_e, \pi_e) - \zeta$, where $\zeta = \frac{4\alpha\epsilon}{(1-\gamma)^2}$. Proof. See Appendix.*

*Proof.* The total variance distance between the ego policy $\pi_e$ and the mixture partner policy policy $\pi_p$ satisfies $D_{TV}(\pi_e(.|s), \pi_p(.|s)) \leq \epsilon$, derived by:

$$
\begin{aligned}
D_{TV}(\pi_e(.|s), \pi_p(.|s)) &= \frac{1}{2} \sum_a |\pi_e(a|s) - \pi_p(a|s)| \\
&= \frac{1}{2} \sum_a |\pi_e(a|s) - (1-\epsilon)\pi_e(a|s) - \epsilon\pi_r(a|s)| \\
&= \frac{1}{2} \sum_a |\epsilon\pi_e(a|s) - \epsilon\pi_r(a|s)| \\
&= \frac{\epsilon}{2} \sum_a |\pi_e(a|s) - \pi_r(a|s)| \\
&\leq \frac{\epsilon}{2} \sum_a (|\pi_e(a|s)| + |\pi_r(a|s)|) \\
&= \frac{\epsilon}{2} \sum_a (\pi_e(a|s) + \pi_r(a|s)) \\
&= \epsilon
\end{aligned}
\tag{12}
$$

Next, following the Lemma 1 and the equivalence of the value estimations as shown in Equation (10), we have that $|V(\pi, \pi_{\mathrm{p}}) - V(\pi, \pi_{\mathrm{e}})| <= \frac{2\alpha\epsilon}{(1-\gamma)^2}, \forall\pi$, where $\delta = \frac{2\alpha\epsilon}{(1-\gamma)^2}$. Combing the definition of $\pi'_{\mathrm{e}} = \arg\max_\pi V(\pi, \pi_{\mathrm{p}})$, we have

$$V(\pi'_{\mathrm{e}}, \pi_{\mathrm{e}}) \geq V(\pi'_{\mathrm{e}}, \pi_{\mathrm{p}}) - \delta \geq V(\pi_{\mathrm{e}}, \pi_{\mathrm{p}}) - \delta \geq V(\pi_{\mathrm{e}}, \pi_{\mathrm{e}}) - 2\delta, \tag{13}$$

let $\zeta = 2\delta$, then the proof is completed. $\square$

## A.4   Proof of Proposition 3

**Proposition 3.** *Denote $\pi_\tau(.|s_t) = \mathbb{E}_{\tau_t}[F(s_t, E_c(\tau_t))]$ as the expectation of the proposed partner prediction module over historical trajectory $\tau_t$. The ground-truth partner policy is denoted as $\pi_p(.|s_t)$. Assume the learned $\pi_\tau(.|s_t)$ satisfies that $D_{TV}(\pi_\tau(.|s_t), \pi_p(.|s_t)) \leq \epsilon_m, \forall s_t$. Define the $\mu^*_\tau$ and $\mu^*_p$ are the best response to the partner prediction module and the ground-truth partner policy respectively. Then the value functions of these best responses collaborating to the partner policy satisfy that: $|V(\mu^*_\tau, \pi_p) - V(\mu^*_p, \pi_p)| \leq \frac{2\gamma\epsilon_m\alpha}{(1-\gamma)^2}$. Proof. See Appendix.*

*Proof.* Recall that, when the partner policy $\pi_p$ is given with $\mathbf{a}^p_t = \pi_p(.|s_t)$, the two-player MDP with transition function $\mathcal{T}$ can be converted in to a single-agent MDP with transition function denoted as $\mathcal{T}_{\pi_p}(s'|s_t, a) = \sum_{a^p} \pi_p(a^p|s_t)\mathcal{T}(s'|s_t, a^p, a)$. The total variance distance between single-agent transition functions under the partner prediction module or the ground-truth partner policy is given by:

$$
\begin{aligned}
D_{TV}\left(\mathcal{T}_{\pi_\tau}(.|s, a), \mathcal{T}_{\pi_p}(.|s, a)\right) &= \frac{1}{2}\sum_{s'}\sum_{a^1}\left|\pi_\tau(a^1|s_t) - \pi_{\mathrm{p}}(a^1|s)\right|\mathcal{T}(s'|s, a^1, a) \\
&\overset{(1)}{=} \frac{1}{2}\sum_{a^1}|\pi(a^1|s) - \pi_{\mathrm{p}}(a^1|s)|\mathcal{T}(s''|s, a^1, a) \\
&= \frac{1}{2}\sum_{a^1}|\pi(a^1|s) - \pi_{\mathrm{p}}(a^1|s)| \\
&= D_{TV}(\pi_\tau(.|s), \pi_{\mathrm{p}}(.|s)) \\
&\leq \epsilon_m
\end{aligned}
\tag{14}
$$

The equation (1) is holds because that the transition function $\mathcal{T}$ is deterministic and equals to 1 at a unique next state $s''$. Define the $\mu^*_\tau$ and $\mu^*_{\mathrm{p}}$ are the best response to the partner prediction module and the ground-truth partner policy respectively. Then the performance gap of these best responses collaborating to the partner policy is bounded by:

$$
\begin{aligned}
&\left|V(\mu^*_\tau, \pi_{\mathrm{p}}) - V(\mu^*_{\mathrm{p}}, \pi_{\mathrm{p}})\right| \\
&= \left|V^{\mathcal{T}_{\pi_{\mathrm{p}}}}(\mu^*_\tau) - V^{\mathcal{T}_{\pi_{\mathrm{p}}}}(\mu^*_{\mathrm{p}})\right| \\
&= \left|V^{\mathcal{T}_{\pi_{\mathrm{p}}}}(\mu^*_\tau) - V^{\mathcal{T}_{\pi_{\mathrm{p}}}}(\mu^*_{\mathrm{p}}) - V^{\mathcal{T}_{\pi_\tau}}(\mu^*_\tau) + V^{\mathcal{T}_{\pi_\tau}}(\mu^*_{\mathrm{p}}) + V^{\mathcal{T}_{\pi_\tau}}(\mu^*_\tau) - V^{\mathcal{T}_{\pi_\tau}}(\mu^*_{\mathrm{p}})\right| \\
&= \left|V^{\mathcal{T}_{\pi_{\mathrm{p}}}}(\mu^*_\tau) - V^{\mathcal{T}_{\pi_\tau}}(\mu^*_\tau) + V^{\mathcal{T}_{\pi_\tau}}(\mu^*_{\mathrm{p}}) - V^{\mathcal{T}_{\pi_{\mathrm{p}}}}(\mu^*_{\mathrm{p}}) + V^{\mathcal{T}_{\pi_\tau}}(\mu^*_\tau) - V^{\mathcal{T}_{\pi_\tau}}(\mu^*_{\mathrm{p}})\right| \\
&= \left|V^{\mathcal{T}_{\pi_{\mathrm{p}}}}(\mu^*_\tau) - V^{\mathcal{T}_{\pi_\tau}}(\mu^*_\tau) + V^{\mathcal{T}_{\pi_\tau}}(\mu^*_{\mathrm{p}}) - V^{\mathcal{T}_{\pi_{\mathrm{p}}}}(\mu^*_{\mathrm{p}}) + \delta\right| \\
&\overset{(1)}{\leq} \left|V^{\mathcal{T}_{\pi_{\mathrm{p}}}}(\mu^*_\tau) - V^{\mathcal{T}_{\pi_\tau}}(\mu^*_\tau) + V^{\mathcal{T}_{\pi_\tau}}(\mu^*_{\mathrm{p}}) - V^{\mathcal{T}_{\pi_{\mathrm{p}}}}(\mu^*_{\mathrm{p}})\right| \\
&\leq \left|V^{\mathcal{T}_{\pi_{\mathrm{p}}}}(\mu^*_\tau) - V^{\mathcal{T}_{\pi_\tau}}(\mu^*_\tau)\right| + \left|V^{\mathcal{T}_{\pi_\tau}}(\mu^*_{\mathrm{p}}) - V^{\mathcal{T}_{\pi_{\mathrm{p}}}}(\mu^*_{\mathrm{p}})\right| \\
&\overset{(2)}{\leq} \frac{2\gamma\epsilon_m\alpha}{(1-\gamma)^2}
\end{aligned}
\tag{15}
$$

Due to the optimality of $\mu^*_{\mathrm{p}}$ when collaborating with the partner policy $\pi_{\mathrm{p}}$, thus $V(\mu^*_\tau, \pi_{\mathrm{p}}) - V(\mu^*_{\mathrm{p}}, \pi_{\mathrm{p}}) \leq 0$. In addition, according to the definitions of $\mu^*_{\mathrm{p}}$, i.e, the optimal policy under the MDP with transition function $\mathcal{T}_{\pi_p}$, and $\mu^*_\tau$, i.e, the optimal policy under the MDP with transition function $\mathcal{T}_{\pi_\tau}$, thus we have $V^{\mathcal{T}_{\pi_{\mathrm{p}}}}(\mu^*_\tau) - V^{\mathcal{T}_{\pi_{\mathrm{p}}}}(\mu^*_{\mathrm{p}}) \leq 0$, $V^{\mathcal{T}_{\pi_\tau}}(\mu^*_{\mathrm{p}}) - V^{\mathcal{T}_{\pi_\tau}}(\mu^*_\tau) \leq 0$. Let $\delta =$

$V^{\mathcal{T}_{\pi_\tau}}(\mu_\tau^*) - V^{\mathcal{T}_{\pi_p}}(\mu_\tau^*) \geq 0$. In summary, the inequality (1) holds because $V(\mu_\tau^*, \pi_p) - V(\mu_p^*, \pi_p) \leq 0$, $V^{\pi_p}(\mu_\tau^*) - V^{\pi_p}(\mu_p^*) \leq 0$, $V^{\pi_\tau}(\mu_p^*) - V^{\pi_\tau}(\mu_\tau^*) \leq 0$ and $\delta \geq 0$.

The inequality (2) holds by applying Lemma 2 with the condition $D_{TV}\left(\mathcal{T}_{\pi_\tau}(.|s,a), \mathcal{T}_{\pi_p}(.|s,a)\right) \leq \epsilon_m, \forall s, a$ satisfied as proven in Equation 14 . $\qquad\square$

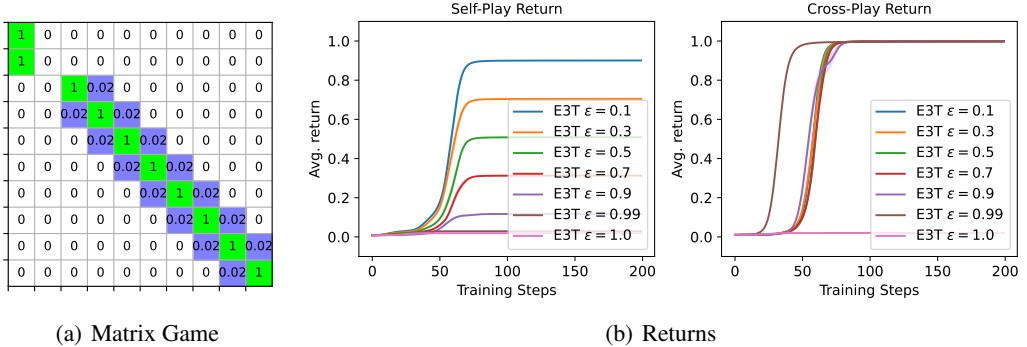

(a) Matrix Game                  (b) Returns

Figure 7: (a) The illustration of $10 \times 10$ sub-matrix of in the upper left of the $100 \times 100$ matrix. (b) The returns of E3T with varying balance parameters $\epsilon$. The left subfigure shows the self-play returns and the right subfigure shows the coordination returns across 10 independently trained policies.

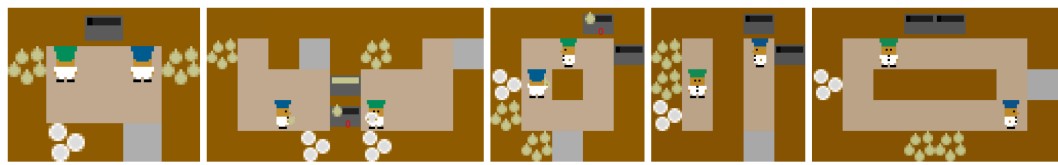

Figure 8: Overcooked Layouts. From left to right: the layouts are *Cramped Room*, *Asymmetric Advantages*, *Coordination Ring*, *Forced Coordination*, and *Counter Circuit*.

## B  Experiment Settings

### B.1  Environments

**A $100 \times 100$ Matrix game**  We verify the effectiveness of the mixed partner policy for learning a robust policy on a one-step matrix game. The matrix is partially shown in Figure 7(a), where we zoom in on the upper left $10 \times 10$ sub-matrix. The matrix denoted as $M$ is constructed such that $M[i,i] = 1$, $M[i+1,i] = M[i,i+1] = 0.2$ for $i \geq 3$, and $M[1,1] = M[2,1] = 1$. All other elements are zero.In the matrix game, two players select a row and a column independently, and the reward is determined by the intersection of their choices. We use a one-step matrix game to test the robustness of our mixture partner policy. The game has a $100 \times 100$ matrix, where each element represents the reward for both players if they choose the corresponding row and column.

**Overcooked**  We validate the mixed partner policy and the partner modeling module for improving zero-shot coordination on the Overcooked environment [8]. In this environment, the target of players is to make soup using onions, and dishes, and then serve it. After successfully serving a soup, players gain 20 rewards. Its discrete action space contains 6 actions: up, down, left, right, noop, and interact.

We conduct experiments on 5 layouts illustrated in Figure 8. The calculation of the cardinality of state spaces shown in Table 1 is explained as follows. Let's take the example of calculating the state space cardinality for the Counter Circuit layout. In the Counter Circuit layout, there are 16 positions on the kitchen counter where plates, onions, soup, or nothing can be placed. Each pot can be in one of 24 states, which includes being empty, having one onion, two onions, three onions, or waiting for

20 counts. There are two players, each of whom can stand at one of 14 grids and face one of four directions. Each player can hold either a plate, an onion, soup, or nothing in their hands. Therefore, the total number of possible states for this layout is: $4^{16}C_{24}^2 24^2 4^2 4^2 \approx 5.8 \times 10^{16}$. Despite having a smaller state space, Forced Coordination is more challenging than Asymmetric Advantages for zero-shot coordination. This is because the two players are separated and must complete different tasks in a timely manner to finish a soup in Forced Coordination, while players can independently complete a soup in Asymmetric Advantages.

## B.2 Baselines

We compare proposed E3T with 6 baselines: Self-Play (SP) [28], Population-Based Training (PBT) [16, 8], Fictious Co-Play (FCP) [32], Trajectory Diversity (TrajeDi) [21], Maximum-Entropy Population-based training (MEP) [42] and Other-Play (OP) [15]. The baselines are described in detail below:

**Self-Play (SP)** [28] In coordination self-play, two players share the same policy during sampling and iteratively improve the policy by focusing on coordinating well with itself. If it converges, the self-play policy is able to coordinate with itself but is poor in generalization when meeting unseen policies.

**Population Based Training (PBT)** [16, 8] PBT is an online evolutionary algorithm, which is widely used in multi-agent RL to iteratively improve policies and finally converge to a population with high quality. At each iteration, paired policies randomly sampled from the population are improved by interacting with a number of timesteps, and then the population is updated according to the recorded performance. Specifically, the worst policy in the population is replaced with a mutation of the best policy.

**Fictitious Co-Play (FCP) [32]** FCP trains a policy that coordinates well with unseen humans by making the best response to a diverse population of partners. To obtain the population, it trains several self-play policies with different random seeds, in addition, it saves past checkpoints with different levels of coordination during training self-play policies.

**Trajectory Diversity (TrajeDi) [21]** TrajeDi solves the two-player zero-shot coordination problem by jointly training a diverse population and its common best response (BR). To be as diverse as possible, the training of the population is regularized by maximizing the Jensen-Shannon divergence (JSD) on the trajectory distributions of policies. Additionally, it trains a common BR to a diverse population, which indicates that the BR is able to coordinate well with unseen policies. We reuse the results of TrajeDi coordinating with human proxies reported in [42].

**Maximum Entropy Population-based training (MEP) [42]** MEP first trains a diverse population through PBT, and this population is encouraged to be diverse with the entropy of the mean policies in the population as an auxiliary reward. Secondly, a common best agent is trained to cooperate with policies in this diversified population, and the selection of partner policies is via prioritized sampling. The prioritization is dynamically updated based on coordination rewards during training progress.

**Other-Play [15]** Both E3T and other-play [15] both belong to the self-play training framework, and we discuss the difference between them as follows. Other-play avoids trapping into specialized conventions by breaking known symmetries in environments, for example the colors in Card games are symmetric, and thus improves the robustness of the trained ego agent. However, this approach relies heavily on the assumption of the state or action space is strongly symmetric, for example, the colors of the cards are symmetric on Hanabi. But it does not adapt to the Overcooked environment, a population two-player cooperative game, since there do not exist obvious and easy to exchange symmetries. Our work shares the benefits of other-play with end-to-end training and breaking the symmetric of self-play training, but E3T is more general since it incents partner exploration by employing a constant random policy and does not rely on any assumptions on environments.

**E3T** We propose an efficient and robust training approach, combing a mixed partner policy for enhancing the diversity of partner policy and the partner modeling module for better adaptation. An ego policy is trained by pairing with the mixture partner which consists of the parameter-copied ego policy and the random policy. We set the length of the context $k = 5$.

## B.3 Experiments design of zero-shot coordination with humans

We describe the details of Human-AI coordination experiments. The results of baselines coordinating with real humans are shown in Figure 4(b). We follow the Human-AI coordination testing protocol proposed in [32]. We recruit 20 individuals to evaluate the coordination ability of E3T and state-

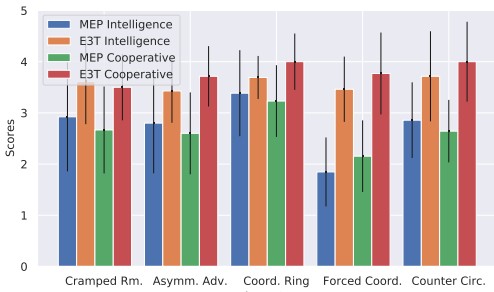

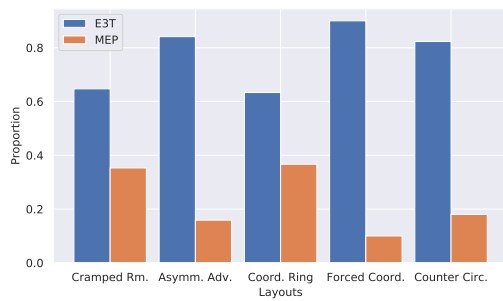

(a) Rating on the intelligence level and coordination ability

(b) Preference for MEP and E3T

Figure 9: Subjective assessments of E3T and MEP providing by real humans. (a). Average scores of the intelligence degree and the coordination ability, both scores are rated in the range of $[1, 5]$, and a larger score indicates higher intelligence and coordination ability. (b). The proportion of real humans' preference for E3T and MEP.

of-the-art MEP, humans play a total of $600$ rounds with them. For cost and time consideration, we reuse the Human-AI coordination results of other baselines reported in [8, 42]. We select models with the best coordination performance with human proxies for both E3T and MEP. For E3T, we set the $\epsilon = 0.5$ on layouts except for *Forced Coord.* with $\epsilon = 0.0$ on it. We use the code provided in [20] to implement the Human-Ai interaction interface.

Participants first watched a description video explaining the game rules and then they were familiar with the experimental process on a single-agent demo layout with the other player not moving to experience the game. During formal experiments, each participant cooperated with the AI agents to complete the 5 layouts and is suggested to get as larger rewards as possible. Pairing with each agent on each layout will repeat 3 times, and the experiment has a total of 30 rounds, with each round lasting one minute. We incentivized game performance: participants earned $50$ yuan to complete $30$ rounds, and the participants with the top $5$ highest rewards earned an extra $50$ yuan.

Besides the objective coordination rewards, we also collected participants' subjective assessment of E3T and MEP about the degree of intelligence, collaboration ability, and preference between the two. After the experiments of each layout, participants were required to answer these three questions. Study sessions lasted around $50$ minutes on average, including being familiar with the experiment process, playing with AI agents, recording rewards, and filling in subjective assessments.

Figure 4(b) shows that E3T outperforms all other baselines when zero-shot coordinating with real humans on average reward across all layouts. Subjective assessments of E3T and MEP are shown in Figure 9. These results show that participants rate E3T with higher scores on the intelligence level and the coordination ability than MEP, across all layouts. In addition, more participants prefer E3T than MEP.

### B.4 The network details of E3T

The network architecture of E3T including the policy network and the partner modeling module is described in Table 4. We describe E3T for the Overcooked environment, which has a discrete action space with 6 actions. E3T are trained based on Tensorflow-GPU 1.15.5 and cuda11.4+cudnn8.8.

## C Additional Results

### C.1 Results on $100 \times 100$ Matrix Game

The performance of E3T under different balance parameters $\epsilon$ is illustrated in Figure 7(b). When $\epsilon = 1.0$, the ego policy is trained by pairing with the completely random partner policy and fails to coordinate during self-play and cross-play with other independently trained ego policies. When $\epsilon < 1.0$, E3T sacrifices some self-play returns during training due to the random policy mixture but

Table 4: The network details of E3T.

| | **Partner Prediction Module** | | |
|---|---|---|---|
| index | Layers | Channels | Kernel Size |
| 1 | Conv2D + LeakyReLU | 25 | 5*5 |
| 2 | Conv2D + LeakyReLU | 25 | 3*3 |
| 3 | Conv2D + LeakyReLU | 25 | 3*3 |
| 4 | Concatenate state embedding and action embedding | / | / |
| 5 | (FC + LeakyRelu)×3 | 64 | / |
| 6 | Concatenate embedding of 5 state-action pairs | 64×5 | / |
| 7 | (FC + LeakyRelu)×3 | 64 | / |
| 8 | FC + Tanh | 64 | / |
| 9 | FC + L2 normalize | 6 | / |
| | **Policy Network** | | |
| index | Layers | Channels | Kernel Size |
| 1 | Conv2D + LeakyReLU | 25 | 5*5 |
| 2 | Conv2D + LeakyReLU | 25 | 3*3 |
| 3 | Conv2D + LeakyReLU | 25 | 3*3 |
| 4 | Concatenate state embedding and action embedding | / | / |
| 5 | (FC + LeakyRelu)×3 | 64 | / |
| 6 | FC | 6 | / |

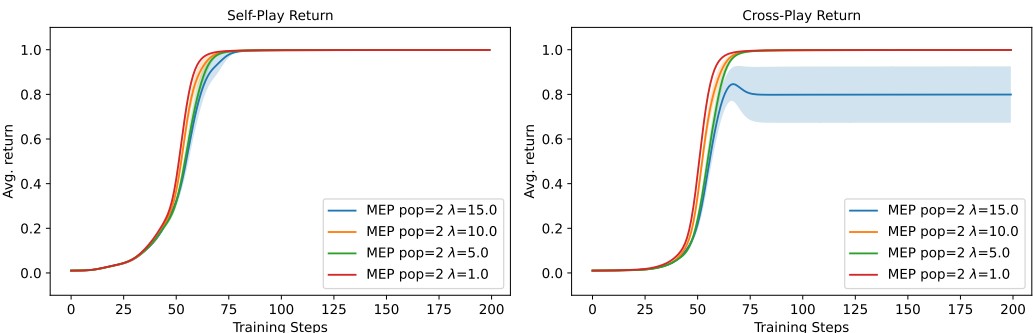

Figure 10: Results of MEP with varying entropy coefficient on $100 \times 100$ matrix game.

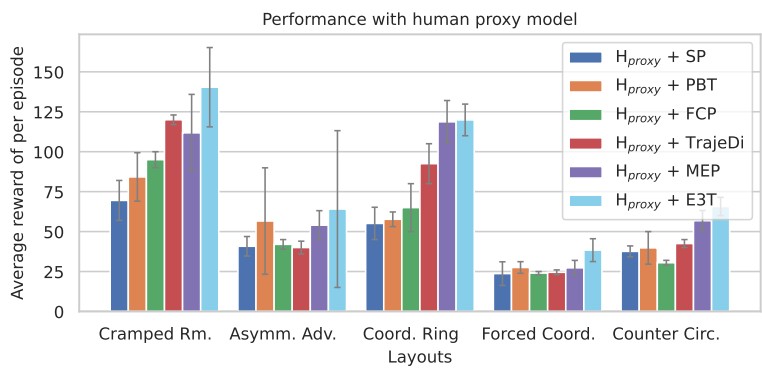

Figure 11: Results on coordinating with human proxy models. The difference between this figure with the Figure 4(a) is that the starting positions of two agents are switched. We plot the mean and standard error of coordination rewards over $5$ random seeds. The parameter $\epsilon$ is set as $0.5$ for all layouts except for *Forced Coord.*, which sets the $\epsilon$ as $0.0$. In this layout, the ego agent does not need too much exploration due to the range of activity being very narrow.

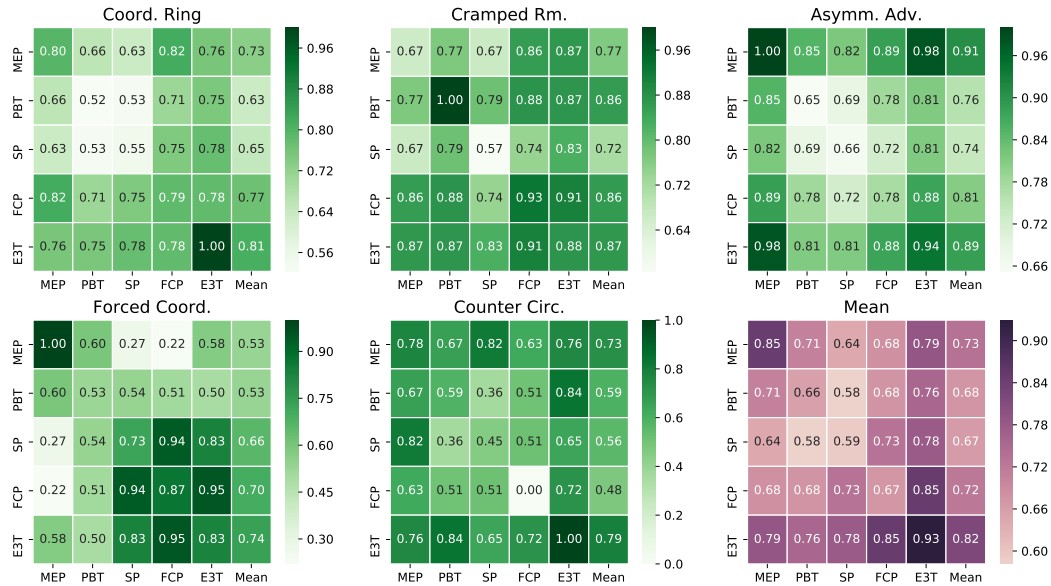

Figure 12: Results on coordinating with other AI baselines. These results plot the normalized cross-play [15] rewards between baselines. The first 5 subfigures show the results on 5 layouts and the last shows the average results over 5 layouts. The last column of each subfigure represents the average result of collaborating with other baselines. For each baseline, we train 5 different models varying only over random seeds. The hyper-parameter $\epsilon$ is set as $0.5$ on *Cramped Rm.* and *Asymm. Adv.*, as $0.3$ on *Coord. Ring*, and *Counter Circ* and a $0.0$ on *Forced Coord.*

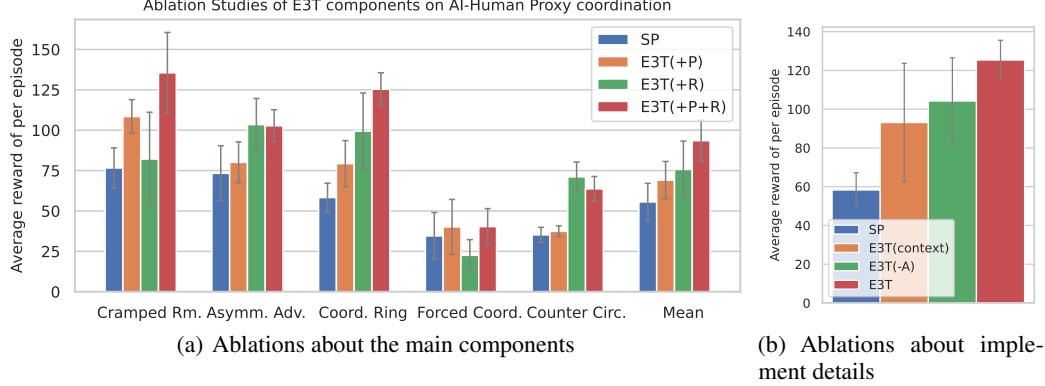

(a) Ablations about the main components

(b) Ablations about implement details

Figure 13: Ablation of E3T's components on collaborating with human proxies. We plot the average performance over standing in two different positions. (a) (+R) and (+P) respectively integrate the mixed partner policy and the partner modeling module into the self-play framework. For versions with the mixed partner policy (+R), the balance parameter $\epsilon$ is defined as $0.5$, and as $0.3$ on *Forced Coord.*. (b) E3T(context) means that the policy depends on the concatenation of the state and the context embedding. E3T(-A) means that both the partner policy and ego policy use outputs of the partner prediction module as additional latent. We plot the results of models coordinating with the human proxy on *Coord. Ring* with $\epsilon = 0.5$.

achieves optimal cross-play returns during testing. These results demonstrate that E3T can explore effectively and avoid being stuck in suboptimal on the diagonal line as shown in Figure 7(a).

Results of MEP with varying population entropy coefficients are shown in Figure 10. The entropy coefficient corresponds the $\lambda$ in Equation (1). MEP with population=2 can reach the best self-play returns but fail to coverage to the optimal policy (with the first column, the first and second rows)

when $\lambda = 15$, which indicates the hyper-parameter $\lambda$ for reward reshaping is hard to tune. We select $\epsilon = 0.99$ and $\lambda = 5.0$ for results in Figure 3(a).

## C.2  Results on Overcooked

Results on collaborating with human proxies with starting position switched are shown in Figure 11. E3T achieves superior or comparable performance compared to other baselines.

**AI-AI coordination results on 5 layouts** Additional results of AI-AI coordination on all layouts and the average performance over all layouts are shown Figure 12. E3T outperforms other baselines on $4$ layouts and has comparable performance with population-based method MEP on *Asymm. Adv.*. In addition, E3T achieves the highest average performance over all layouts when collaborating with AI baselines. Moreover, MEP and FCP training the ego agent by pairing with pre-trained diverse partner populations are better than SP and PBT, which only improve the coordination rewards during training.

**Ablation studies of E3T's components on cooperation with human proxies** The ablation studies of E3T's components on cooperation with human proxies are shown in Figure 13(a). The average reward of E3T(+P) is larger than that of SP on each layout, thus the partner modeling can assist the zero-shot coordination since the ego policy adapts to the predicted partner actions. In addition, E3T(+R) also outperforms SP on $4$ layouts except for the *Forced Coord.*, which indicates that encouraging the partner policy to be more diverse during training time is beneficial to zero-shot coordinating with unseen partners. Moreover, E3T(R+P) that combing the partner modeling and the mixed policy is better than other models on the averaged reward over all layouts. Figure 13(b) compares the ego policy that conditions on the context embedding with the one that conditions on the predicted partner action. The results show that the latter policy performs better, suggesting that the predicted partner action is easier to comprehend and respond to than the context embedding. Moreover, E3T(-A) has lower coordination performance than E3T, which demonstrates the advantage of using asymmetric latent variables for ego and partner policies during training, as explained in section 4.3. This technique enhances zero-shot coordination by introducing more diversity in the latent space and breaking the asymmetry between ego and partner policies.

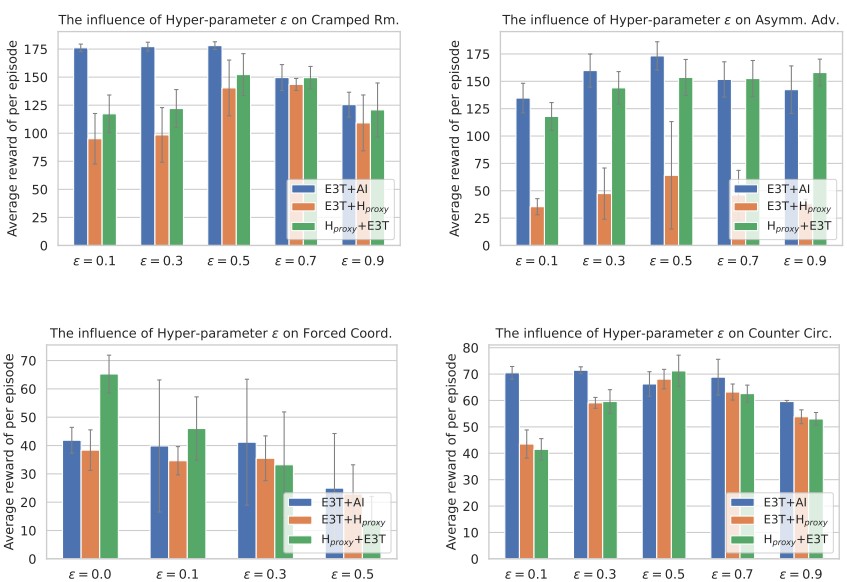

Figure 14: Ablation studies of $\epsilon$ on 4 layouts. In general, the $\epsilon$ can be set small when the layout space is not large, e.g. smaller $\epsilon$ for the Forced Coordination layout. For layouts with big space, a larger $\epsilon$ can provide more explanation about different cooperation patterns.

**Further ablation studies of $\epsilon$ on more layouts** Figure 14 Ablation studies of epsilon (in the range of [0.1, 0.3, 0.5, 0.7, 0.9]) on the other 4 layouts are shown in Figure 14 of the rebuttal material. In general, the epsilon can be set small when the layout space is not large, e.g. smaller epsilon

for the Forced Coordination layout can achieve better performance because the activation range is narrow and the partner's coordination ability is more important than behavior diversity in this case. For layouts with big spaces, there exist different coordination patterns, such as multiple paths to deliver soups. Therefore, we normally set the epsilon larger to encourage the policy to explore more coordination patterns. For example, our method achieves higher rewards with epsilon=0.5 when coordinating with human proxies in other layouts, and with epsilon=0.3 or 0.5 when coordinating with other AI baselines.

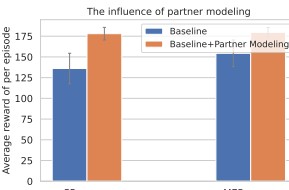 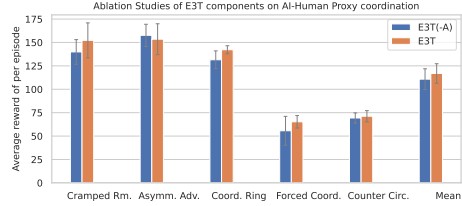

(a) Effect of partner modeling      (b) Effect of asymmetric action distribution

Figure 15: (a) We plot the effect of the partner modeling to Self-play and MEP when collaborating with AI baselines. (b) We plot the effect of the asymmetric input setting. E3T(-A) indicates both ego and partner policies are equipped with the partner modeling module.

**The effect of the partner modeling module** We have incorporated the partner modeling module into two baselines (Self-play and the state-of-the-art MEP) as illustrated in Figure 15(a). These results demonstrate that the partner modeling module can further enhance the zero-shot coordination performance of baselines when they collaborate with AI baselines.

**The effect of the asymmetric inputs for the ego and partner policies** This work makes use of predicted partner action distribution for ego policy and random action distribution for the partner policy during training. The motivation of the asymmetric input setting comes from two considerations. Firstly, the random action distribution introduces extra randomness into the partner policy, which in turn improves partner diversity. Secondly, this is consistent with the real-world command that AI agents should adapt to human partners' behavior, while different humans may have various cooperative patterns and are not necessary to try to fit AI agents.

In addition, we have empirically verified that asymmetric input is helpful to zero-shot coordination as illustrated in Figure 15(b) of the rebuttal material, where the performance of our method with asymmetric input slightly outperforms the model that uses the partner modeling for both ego and partner policies on most layouts.

**The partner modeling module with only the historical observations accessible** In more complex environments, we do not have the ground truth of partner actions, but only the historical observations of the ego policy are available, thus it is worth developing an advanced model to predict the partner action from changes in the observed environment. In the Overcooked environment, agents can observe the global state (this is the environmental setting). We also empirically verified that partner action can be predicted from the sequence of ego states on Coord. Ring layout. As shown in Table 5, the prediction accuracy from only the state sequence is comparable to that from the state-action sequence. Note that the accuracy of random prediction is around 0.17.

Table 5: The accuracy of partner prediction and the coordination performance under the settings of the state or state-action available.

|  | Prediction accuracy | Coordination reward with human proxy |
|---|---|---|
| state-action | $0.54 \pm 0.04$ | $143.3 \pm 8.3$ |
| state | $0.50 \pm 0.01$ | $139.3 \pm 6.8$ |

