# OpenReview forum: "An Efficient End-to-End Training Approach for Zero-Shot Human-AI Coordination"
_NeurIPS.cc/2023/Conference — NeurIPS 2023 poster_

### Official Review · Reviewer_DWEk · 2023-07-06

**Soundness:** 3 good
**Presentation:** 4 excellent
**Contribution:** 3 good
**Rating:** 7
**Confidence:** 4

**Summary:**

The authors propose an Efficient End-to-End training (E3T) approach for zero-shot human-AI coordination. E3T uses a mixture of an ego policy and a random policy as a simple way of training a diverse policy that is still capable of coordination. Unlike prior population based approaches, E3T does not require training populations of agents and thus is more efficient. The authors also propose a partnering module that models the agent’s partner’s actions, enabling improved zero-shot human collaboration. The experiments show a clear improvement over a range of prior methods, testing with both proxy and real human partners.

**Strengths:**

- The idea is straightforward and easy to employ, and directly addresses a meaningful challenge (efficient zero-shot coordination with humans). The combination of an ego policy and a random policy brings together the strengths of self play for training a coordination policy, whilst also incorporating diversity through the random policy to prevent overfitting to a specific partner.
-  The inclusion of a partnering module makes sense and shows a clear advantage empirically.
- The experiments are thorough and show comparisons against a range of prior approaches, including experiments with real people, as well as detailed ablations of the proposed method.

**Weaknesses:**

- The method seems like it would be sensitive to $\varepsilon$. While 5b) shows that evaluation for one task, it would be interesting to see of all tasks (i.e. do we need to tune $\varepsilon $ for each task separately?)
- nit: missing epsilon in line 7 in the algorithm
- nit: line 232 has a typo — "Assume the *learned*..."

**Questions:**

Based on figure 6, the overall accuracy of action predictions is pretty low (less than 60%) — it’s surprising that the partnering module is still helpful even when accuracy is not very high. Why do the authors think this is?

**Limitations:**

Limitations are clearly discussed in the paper.

---

> ### Author Rebuttal · Authors · 2023-08-10
>
> We thank the time and effort reviewer DWEk has invested in reviewing our paper, and we appreciate that you concur with the main advantages of our method: (1) simplicity of the method (2) the reasonableness and effectiveness of introducing the partner modeling module (2) thorough experiments and superior performance over existing methods. We have provided detailed explanations and clarifications to resolve your concerns regarding experiments, and thus respectfully hope you can consider the response to the final decision.
>
> > Q1: The method seems like it would be sensitive to . While 5b) shows that evaluation for one task, it would be interesting to see of all tasks (i.e. do we need to tune  for each task separately?)
>
> Thank you for your question.  Ablation studies of epsilon (in the range of [0.1, 0.3, 0.5, 0.7, 0.9]) on the other 4 layouts are shown in Figure 2. of the rebuttal material. In general, the epsilon can be set small when the layout space is not large, e.g. smaller epsilon for the Forced Coordination layout can achieve better performance because the activation range is narrow and the partner's coordination ability is more important than behavior diversity in this case. For layouts with big space, we normally set the epsilon larger. For example, our method achieves higher rewards with epsilon=0.5 when coordinating with human proxies in other layouts, and with epsilon=0.3 or 0.5 when coordinating with other AI baselines.
>
> > Q2: nit: missing epsilon in line 7 in the algorithm
> >
> > nit: line 232 has a typo — "Assume the *learned*..."
>
> Thanks for pointing out these typos, and we will correct them in the revision following your suggestions.
>
> > Q3: Based on figure 6, the overall accuracy of action predictions is pretty low (less than 60%) — it’s surprising that the partnering module is still helpful even when accuracy is not very high. Why do the authors think this is?
>
> Thank you for your question. Although the partner modeling module can not exactly predict the partner's action(from 6 actions), it has a relatively high prediction accuracy (less than 60%) compared to a random prediction (with a probability of 1/6, less than 20%). Therefore, we think the partner modeling module can potentially reason about the behavior patterns of unseen partners. Figure 6 shows that coordination performance improves as prediction accuracy increases.

---

> > ### Comment · Reviewer_DWEk · 2023-08-14
> >
> > Thank you for the response and additional ablations. I think adding those experiments and the clarifications to my questions above to the draft will strengthen the paper. I retain my original score.

---

### Official Review · Reviewer_dKj6 · 2023-07-06

**Soundness:** 3 good
**Presentation:** 4 excellent
**Contribution:** 3 good
**Rating:** 7
**Confidence:** 4

**Summary:**

This paper proposes E3T, an method to train agent for zero-shot coordination with humans. The main contribution in E3T is that, in contrast to population-based approaches, it can be trained in a single stage, significantly reducing training time. E3T also includes a model to predict the next action of the partner agents, given that trajectory. By conditioning on that action, the coordination agent can adapt to different partner behaviors according to their observed trajectories. The method is tested on Overcooked against both zero-shot coordination agents and real humans, showing improved performance and lower training time than competitive zero-shot coordination methods.

**Strengths:**

Originality
- The proposed method is a simple extension to MEP, with the adition of the partner action prediction module, which have been explored before. Despite highly relying on these 2 components the method is simple and provides gains in training time and performance.

Quality:
- The paper provides thorough experiments showing that the proposed method can coordinate with several zero-shot coordination agents in different tasks and layouts. The proposed method shows better training time than competitive approaches, as well as improved coordination results. More importantly, it shows significant improvements over baselines when coordinating with real humans.

- Clear and simple to implement method, with comparisons with main coordination approaches and analysis of the entropy and reward bounds of the proposed method.


Clarity:
- The paper provides a clear overview of the main methods for zero-shot coordination and their limitations. The proposed method is clearly explained, and section 4.1 provides a clear intution for the design choices, building from the limitations of self-play appraoches such as MEP to motivate different design decisions.

- The explanation of the algorithm, figures and pseudocode provide a clear udnerstanding of the proposed method.

- Clear analysis of the results, and ablations to understand the importance of the two components


Significance:
- Building agents that can coordinate with other agents in a zero-shot manner is an important problem, and as the paper points out, the two stage training of existing approaches make learning highly inefficient. Proposing a zero-shot coordination coordination agent that can be trained in a single stage is an important problem. Overcooked is a very simplified coordination setting, but one that showcases some of the challenges in zero-shot coordination, and thus showing improved performance and training time resuls is a significant result.

**Weaknesses:**

- Figure 5.b seems to indicate that the value of epsilon has a strong effect on the final performance and changes for different kinds of tasks, making it potentially hard to find a good parameter for novel tasks. How is that epsilon chosen in the other experiments?

- Figure 5.a shows that partner modeling has a significant effect in the coordination performance. Given the importance of that module, which is compatible with the other proposed baselines, it would be worth testing whether adding that into them would improve their performance results, even if training time would still be higher.

- The mixture policy looks a lot like a self-play policy where a temperature to add noise to the policy is added. What are the main differences to that, if the partner modeling module is eliminated?

- Propositions 1 and 2 are sound, and show that both the entropy and performance of the mixture policy are lower bounded by random and ego policy counterparts, safe for some terms depending on the mixture parameter and action space size. Despite the analysis being sound I am not sure about the value they add that was not known before. The entropy is equal to that of the random policy when \eps is 1, and decreases as we decrease \eps (increasing C_1). Similar analysis can be done for Proposition 2, meaning that the ego and random policies serve as upper bounds. How do propositions 1 and 2 help?

- Having access to the partners action and state pairs is a big assumption that may not hold in more complex environments.

- It seems strange that an ego policy that is trained with actions coming from the partner action prediction model can generalize when training the partner, where a random action distribution is used at the input. As epsilon becomes smaller, this gap should increase further. Could authors comment on that?

- Related work: Authors should consider citing, and commenting on the differences with https://arxiv.org/pdf/2104.07750.pdf, which also models the partner for multi-agent collaboration.

**Questions:**

- I am surprised at Fig 4.a results, particularly on th efact that E3T coordinates better with self-play than self-play itself. This seems really counterintuitive, could authors comment on that?

**Limitations:**

Yes

---

> ### Author Rebuttal · Authors · 2023-08-10
>
> We thank the time and effort reviewer dKj6 has invested in reviewing our paper, and we appreciate that you concur with the main advantages of our method: (1) simplicity and high training efficiency (2) thorough experiments and superior performance over existing methods (3) clarity of motivation and presentation. We have provided detailed explanations and clarifications to resolve your concerns regarding experiments, and thus respectfully hope you can consider the response to the final decision.
>
> > Q1: How is epsilon chosen in the other experiments?
>
> Thank you for your question. Ablation studies of epsilon on the other 4 layouts are shown in Figure 2. of the rebuttal material. For more details and analysis about the epsilon selection, please see our response to Q1 of reviewer DWEk.
>
> > Q2: About adding partner modeling into other baselines
>
> Thank you for your suggestion. We have incorporated the partner modeling module into two baselines (Self-play and the state-of-the-art MEP) as illustrated in Figure 3. (a) of the rebuttal material. These results demonstrate that the partner modeling module can further enhance the zero-shot coordination performance of baselines when they collaborate with AI baselines.
>
> > Q3: The mixture policy looks a lot like a self-play policy where a temperature to add noise to the policy is added. What are the main differences to that, ......
>
> Thank you for your question. In the implementation, the mixture partner policy $\pi_p=\epsilon\pi_r +(1-\epsilon) \pi_e$ is constructed by adding noise $\pi_r$ (a uniform distribution) to the self-play policy $\pi_e$ with a temperature $\epsilon$. Then, the partner action is sampled from this mixed distribution. This method is inspired by the idea of population-based methods that partner policies should be both skilled in coordination and diverse behaviors. Our method simplifies previous population-based methods as a single-staged framework and an end-to-end training approach, so it can improve training efficiency by 9x compared to the state-of-the-art MEP.
>
> > Q4: Propositions 1 and 2 are sound, ...... How do propositions 1 and 2 help?
>
> Thank you for acknowledging the soundness of our propositions. Intuitively, the diversity reflected by entropy increases, and the coordination performance decreases with the random coefficient epsilon increasing. We attempt to quantitatively verify these intuitions as Propositions 1 and 2. Regarding the $\epsilon$ selection, we rely on empirical analysis to balance the coordination ability and behavior diversity as explained in Q1.
>
> > Q5: Having access to the partners action and state pairs is a big assumption that may not hold in more complex environments.
>
> In more complex environments, we do not have the ground truth of partner actions, but only the historical observations of the ego policy are available, thus it is worth developing an advanced model to predict the partner action from changes in the observed environment. In the Overcooked environment, agents can observe the global state (this is the environmental setting). We also empirically verified that partner action can be predicted from the sequence of ego states on Coord. Ring layout. As shown in the table below, the prediction accuracy from only the state sequence is comparable to that from the state-action sequence. Note that the accuracy of random prediction is around 0.17.
> ||Prediction accuracy|Coordination reward with human proxy|
> |-|-|-|
> |state-action| 0.54(0.04)|143.3(8.3)|
> |state| 0.50(0.01) |139.3(6.8)|
> > Q6: It seems strange that an ego policy that is trained with actions coming from the partner action prediction model can generalize when training the partner, where a random action distribution is used at the input. ......
>
> The motivation of the asymmetric input setting, with predicted partner action distribution for ego policy and random action distribution for the partner policy during training, comes from two considerations. Firstly, the random action distribution introduces extra randomness into the partner policy, which in turn improves partner diversity. Secondly, this is consistent with the real-world command that AI agents should adapt to human partners' behavior, while different humans may have various cooperative patterns and are not necessary to try to fit AI agents.
>
> In addition, we have empirically verified that asymmetric input is helpful to zero-shot coordination as illustrated in Figure 3. (b) of the rebuttal material, where the performance of our method with asymmetric input slightly outperforms the model that uses the partner modeling for both ego and partner policies on most layouts.
>
> > Q7: About differences with the mentioned related work
>
> Thank you for your recommendation. Our work and this paper both infer agents' intention to enhance multi-agent coordination. The intention inference purposes are different in the two works. This paper infers each agent's visual attention region and designs additional rewards to incentive all agents to focus on the same elements of the environment to reduce the cost of multi-agent exploration. Our work infers partner actions from historical contexts, which helps to adapt well to unseen partners with different coordination patterns by reflecting on predicted partner actions. We will include this work in the related work of the revision.
>
> > Q8: About E3T coordinates better with self-play than self-play itself
>
> Thanks. I think you are referring to the coordination results between baselines shown in Fig 4. b of the main paper, where each baseline has 5 different random seeds. Self-play policies may fall into some specific conventions during training and thus fail to cooperate with other self-play policies trained **by different random seeds** during testing. By contrast, E3T with the mixture partner policy and the partner modeling module can adapt well to unseen self-play policies, so it is reasonable E3T can achieve good performance when collaborating with self-play.

---

> > ### Comment · Reviewer_dKj6 · 2023-08-18
> > **Response**
> >
> > Thanks for the rebuttal. It addressed all my questions. Moreover, the experiments where only the state is shown relax the assumptions in the original paper, and still offer high coordination rewards. I am thus changing my rating to accept.

---

### Official Review · Reviewer_kGQD · 2023-07-07

**Soundness:** 3 good
**Presentation:** 3 good
**Contribution:** 2 fair
**Rating:** 6
**Confidence:** 4

**Summary:**

This paper proposes a simple end-to-end training mechanism for zero-shot coordination. Existing works in the ZSC literature often make use of population-based training, where a large pool of policies is trained to play well against itself while maintaining a certain form of diversity to prepare for an unknown online partner. This work proposes that sufficient diversity can be achieved by simply adding epsilon-greedy noise into the self-play policy. Performance is further improved by an additional partner modeling module, which takes the past trajectory of the partner and predicts its next action. Experiments against human players and various baselines in the Overcooked environment show some improvements.

**Strengths:**

- Simplicity of method, where only a single policy is learned. This is much more efficient than population-based training approaches.
- Experiments are performed against various agents, including human proxy, actual human players, and some baselines.
- Code is available in the submission.

**Weaknesses:**

- The limited novelty. The method is a simple combination of existing techniques (epsilon-greedy, partner modeling, etc). Besides ablations, a more detailed analysis would provide more insight as to why this works.
- The Overcooked environments are out-of-date. It is required to further evaluate the method on more complex Overcooked environments[1], e.g.,  more recipes,  and more layouts.

[1 ]Wu, Sarah A., et al. "Too Many Cooks: Bayesian Inference for Coordinating Multi‐Agent Collaboration." Topics in Cognitive Science 13.2 (2021): 414-432.

**Questions:**

-  Are there any details that differ from those in [2], e.g. the human proxy partner or the reward function?
- Can the proposed methods be used in more challenging scenarios, e.g., more players, mixed games, or partial observation?

[2] Strouse, D., McKee, K., Botvinick, M., Hughes, E., and Everett, R. Collaborating with humans without human data. Advances in Neural Information Processing Systems, 34: 14502–14515, 2021.

**Limitations:**

This paper will be improved by evaluating on more complex environments and showing the scalability of the proposed methods.

---

> ### Author Rebuttal · Authors · 2023-08-10
>
> We thank the time and effort reviewer kGQD has invested in reviewing our paper, and we appreciate that you concur with the main advantages of our method: (1) simplicity and high training efficiency (2) experiments are performed against various agents. We have provided detailed explanations and clarifications to resolve your concerns regarding the insights and experiments, and thus respectfully hope you can consider the response to the final decision.
>
> > Q1: The limited novelty. The method is a simple combination of existing techniques (epsilon-greedy, partner modeling, etc). Besides ablations, a more detailed analysis would provide more insight as to why this works.
>
> Thank you for acknowledging the simplicity of our method, which indeed contributes to its higher training efficiency compared to other approaches. However, we'd like to respectively clarify that our method is more than just a straightforward combination of existing techniques. **In fact, we are motivated by the idea of population-based methods that partner policies should be both skilled in coordination and diverse behaviors**. To achieve this, we are the first to decompose the population training objectives of MEP into two policies: one being a simple copy of the ego policy for coordination skills, and the other being a random policy for diverse behaviors.
>
> Although this decomposition appears similar to epsilon-greedy, the underlying motivation and induction process are entirely distinct, and we believe our decomposition is experimentally grounded as results show and the simplicity may benefit or inspire more subsequent works based on population-based methods. Besides ablations, we have also provided a theoretical analysis of partner modeling in Proposition 3, which indicates that smaller partner action prediction error leads to better coordination performance.
>
> > Q2:The Overcooked environments are out-of-date. It is required to further evaluate the method on more complex Overcooked environments[1], e.g., more recipes, and more layouts.
>
> Thank you for sharing this work with us. Although our paper and [1] both conduct experiments on Overcooked, our contributions differ in their focus. Our paper aims to achieve human-AI zero-shot coordination **without using human data,** while [1] whereas proposes a hierarchical planning method that can quickly complete a recipe. Due to the baselines of [1] being based on model-based dynamic programming, they do not support human-AI zero-shot coordination.
>
> We'd like to try to run our method in the environment you suggest, but it is challenging to transfer our codebase and compare zero-shot coordination baselines on the new Overcooked environments[1] within a limited time. In our main paper, we use the same Overcooked environment as human-AI zero-shot coordination baselines (including BCP[3], FCP[2] you mentioned, and MEP), and we have tested our approach on 5 different layouts, which cover the three open, partially passable, and challenging forced coordination scenarios in [1]. These layouts are not as simple as imagined. Here are some analyses of the complexity of these layouts.
>
> |  | *Cramped Room* | *Asymmetric Advantages* | *Coordination Ring* | *Forced Coordination* | *Counter Circuit*   |
> | -- | -- | -- | -- | --- | -- |
> | State space | $5.3 \times 10^7$ | $1.2\times 10^{14}$     | $6.8\times 10^{10}$ | $2.0\times 10^9$  | $5.8\times 10^{16}$ |
>
> With consistently superior performance than baselines in these layouts in the Overcooked environment, we believe our evaluations are sufficient, and we also would like to add the discussion and differences with [1] in the final version of our paper, as per your suggestion.
>
> To further verify the effectiveness of our method and to explore new coordination challenges beyond Overcooked, we alternatively evaluate our method on the Google Football environment following your suggestions. The details and the results are shown in the R1 of our "global" response above.
>
> [1 ]Wu, Sarah A., et al. "Too Many Cooks: Bayesian Inference for Coordinating Multi‐Agent Collaboration." Topics in Cognitive Science 13.2 (2021): 414-432.
>
> [2] Strouse, D., McKee, K., Botvinick, M., Hughes, E., and Everett, R. Collaborating with humans without human data. Advances in Neural Information Processing Systems, 34: 14502–14515, 2021.
>
> [3] Carroll, M., Shah, R., Ho, M. K., Griffiths, T., Seshia, S., Abbeel, P., and Dragan, A. On the utility of learning about humans for human-ai coordination. *Advances in neural information processing systems*, 32, 2019.
>
> > Q3: Are there any details that differ from those in [2], e.g. the human proxy partner or the reward function?
>
> Thank you for your question. We follow the same overcooked environment and human proxies as BCP[3]. The reward function of our work is the same as FCP[2], which gives 20 points for each successful delivery. We also use the same layouts and recipes as FCP. The state representations and the training data for human proxies of FCP[2] are different from that of BCP and our work, so we implemented the FCP[2] based on our codebase for a fair comparison with our method. We would like to publish the implementation in the future.
>
> > Q4:Can the proposed methods be used in more challenging scenarios, e.g., more players, mixed games, or partial observation?
>
> Following your suggestion, we conduct additional experiments on a three-player cooperative game of the Google Football environment. The details and the results are shown in the R1 of our "global" response above.
>
> In summary, we provided additional experiments on a more complex environment following your suggestions and explain the motivation of our method and the difference from the previous work. We believe our response sufficiently addresses your concerns regarding the experiments and novelty, and thus respectfully hope you can reconsider your final decision. If you have further concerns, please feel free to respond to us and we would like to discuss them with you.

---

> ### Author Response · Authors · 2023-08-20
> **Providing additional experiment results on the Google Football game**
>
> Dear Reviewer kGQD,
>
> We really thank you for your valuable comments on improving our work. Here, we would like to add more experiment results and a more detailed analysis of a 3-player Google Football game.
>
> As per your suggestion, we have conducted additional experiments on the Google Football environment's "3 vs 1 with Keeper" layout, which is a three-player cooperative game with 19 actions in the discrete action space. The three players have shared rewards and they cooperate to gain high rewards. We train policies via self-play and our methods. To test the zero-shot coordination ability of these models, we follow the experimental setting of Other-Play [1]. In this setting, **the policies cooperate with unseen teammate policies that were trained with the same algorithms but different random seeds, without any prior interaction.** This ensures that the policies have not encountered their teammates during training. In the table below, we report the mean and standard error of the winning rates of scoring a goal.
>
> |                                       | Training performance | Test performance(Zero-shot coordination  across different random seeds) |
> | ------------------------------------- | -------------------- | ------------------------------------------------------------ |
> | Self-Play                             | 0.87(0.05)           | 0.03(0.01)                                                   |
> | E3T(Mixture Policy)                   | 0.79(0.04)           | 0.70(0.01)                                                   |
> | E3T(Partner Modeling)                  | 0.89(0.05)           | 0.24(0.05)                                                   |
> | E3T(Mixture Policy + Partner Modeling) | 0.87(0.02)           | 0.84(0.02)                                                   |
>
> As shown in the table above,  policies trained by the self-play fail to cooperate with policies from different training seeds, because self-play policies may fall into some specific coordination conventions during training. Our proposed mixture policy can increase the behavioral diversity of teammates, allowing the ego policy to encounter different coordination patterns during training. Consequently, our method with the mixture policy can train a policy that can adapt well to policies independently trained from different training seeds. Moreover, the partner modeling module can further enhance the zero-shot coordination performance by enabling the ego policy to respond to the predicted teammates' action distributions. We note that we set the epsilon as 0.1 for this environment (a large epsilon, i.e., more randomness, would result in policies failing to obtain rewards on this task) and the partner modeling module uses the historical observations of the ego policy to predict the action distributions of all 3 players.
>
> In summary, our method combining the mixture partner policy and partner(teammate) modeling module can also improve the zero-shot coordination ability in the complex football environment, with 3 cooperative players and a large action space with 19 discrete actions in it.
>
>  [1] Hu, H., Lerer, A., Peysakhovich, A., and Foerster, J. “other-play” for zero-shot coordination. In International Conference on Machine Learning, pp. 4399–4410. PMLR, 2020.
>
> We hope that our responses have solved your concerns and respectfully hope you can consider the final decision accordingly. As the author-reviewer discussion period is coming to an end, we are respectfully looking forward to discussing with you. Please let us know if you have any further questions or concerns and we are very happy to address them.

---

> > ### Comment · Reviewer_kGQD · 2023-08-20
> > **Response to Authors**
> >
> > Thanks for your detailed response. After reading your response, most of my concerns are addressed. So I decide to raise my score. Particularly, the results on Google Football are impressive to me. But I have another question about the experiments, how to get the keeper policy? Are they different between testing and training?

---

> > > ### Author Response · Authors · 2023-08-21
> > > **About the keeper policy**
> > >
> > > Thank you for your question. The layout '3 vs 1 with Keeper' we used consists of three players and one keeper. The keeper policy is a built-in bot provided by the Google Football environment, and the keeper policy is the same during the training and testing phases.

---

### Official Review · Reviewer_dkSb · 2023-07-09

**Soundness:** 3 good
**Presentation:** 3 good
**Contribution:** 3 good
**Rating:** 7
**Confidence:** 3

**Summary:**

This paper introduces a one-stage training framework for human-AI coordination. The key insight is that partner policies should exhibit both coordination skills and diversity. However, the traditional approach of constructing a competent and diverse partner population in the first stage and training an ego policy given this population in the second stage is complex and inefficient. To address this, the author proposes to decouple the partner policy into two components: one aligned with self-play (ego policy) and another component introducing diversity (random policy). By implementing this approach, the paper significantly improves efficiency compared to the Maximum Entropy Population (MEP) method. The effectiveness of the proposed method is demonstrated through experiments conducted on five different Overcooked layout scenarios.

**Strengths:**

1. The paper is well-motivated and easy to follow.
2. The key idea behind the proposed method is straightforward and intuitive.
3. The experiments section of the paper is substantial, covering a range of important aspects such as zero-shot coordination with human proxy, comparisons against human and AI baselines, and a well-conducted ablation study.
4. This paper offers a sound theoretical analysis of the proposed method.

**Weaknesses:**

Based on Figure 5, it appears that the proposed method is highly task-sensitive. Therefore, I find it less convincing to solely rely on experiments conducted on five different layouts of Overcooked. It would be beneficial for the authors to consider conducting experiments on various multi-agent reinforcement learning benchmarks, such as matrix games, MPE, Hanabi, SMAC, and others, to provide a more comprehensive evaluation of their method.

**Questions:**

1. The partner policy observes random actions, which implies it is unlikely to exhibit coordinated behavior. And the ego policy achieves the best performance when $\epsilon = 0.5$, which is a considerably large value. Does the ego policy adopt a more conservative approach and attempt to complete the task independently?
2. The significant performance improvement of this method compared to MEP is intriguing. It appears that this paper can be viewed as a one-staged MEP combined with partner modeling. Is the performance disparity primarily attributed to the partner modeling?
3. In the ablation study, when partner modeling is removed, what serves as the input for the ego policy?
4. Minor comments:
a) Figure 6 is difficult to comprehend. The importance of "environment steps" is not clear in the figure. I think it would be helpful to use the "accuracy of action prediction" as the x-axis, as it could better illustrate the correlation between "average reward per episode" and "action prediction".

**Limitations:**

I appreciate that the author acknowledges the limitation of the current method in neglecting general multi-player human-AI coordination tasks.

---

> ### Author Rebuttal · Authors · 2023-08-10
>
> We thank the time and effort reviewer dkSb has invested in reviewing our paper, and we appreciate that you concur with the main advantages of our method: (1) it is well-motivated and easy to follow (2) the substantial experiments (3) sound theoretical analysis of the proposed method. We have provided detailed explanations and clarifications to resolve your concerns regarding the experiments, and thus respectfully hope you can consider the response to the final decision.
>
> > Q1:  ....... It would be beneficial for the authors to consider conducting experiments on various multi-agent reinforcement learning benchmarks, such as matrix games, MPE, Hanabi, SMAC, and others, to provide a more comprehensive evaluation of their method.
>
> Thanks for your suggestion. We point out that we have conducted experiments on a 100x100 matrix game as shown in Figure 4 (a) of the main paper. Our method can achieve the best performance with superior training efficiency than existing baselines. Ablation studies about the random coefficient epsilon are shown in Figure 10 (b) in the appendix, which demonstrate that E3T with $0<\epsilon<1$ can effectively explore the strategy space to find the optimal strategy and to avoid getting trapped in suboptimal solutions.
>
> We conduct additional experiments on the Google Football environment's "3 vs 1 with Keeper" layout.  We provide more details and results in the R1 of our "global" response above.
>
>
> > Q2:The partner policy observes random actions, which implies it is unlikely to exhibit coordinated behavior. And the ego policy achieves the best performance when , which is a considerably large value. Does the ego policy adopt a more conservative approach and attempt to complete the task independently?
>
> Thanks for your question. If without partner modelling, the policy tends to complete the task independently. With partner modeling,  the ego policy adapts to diverse partner behaviors well during training to improve the generalization ability. The design of partner policy merging random actions makes the partner policy more diverse in its behaviors while the ego policy can adjust well to different partner behaviors. This is also consistent with the real-world scenario that the ideal AI agent should adapt to human partners' behavior, while different humans may appear in various cooperative patterns and are not necessary to try to fit AI agents.
>
> > Q3: The significant performance improvement of this method compared to MEP is intriguing. It appears that this paper can be viewed as a one-staged MEP combined with partner modeling. Is the performance disparity primarily attributed to the partner modeling?
>
> The performance improvement of our method attributes to both the partner modeling and the mixture partner policy. The partner modeling benefits zero-shot coordination since the ego policy can respond to the predicted partner action. The mixture partner policy exhibits behavior diversity and coordination competence by directly combing the ego and random policies. In addition, our method is a single-staged framework and an end-to-end training approach, so it can improve training efficiency by 9x compared to the population-based MEP.
>
> We plot the performance of E3T variants and MEP in coordination with AI baselines as shown in Figure 1.(b) of the rebuttal material. These results show that E3T only with the mixture partner policy can achieve comparable performance with MEP, and the partner modeling module can further improve the coordination ability.
>
> > Q4: In the ablation study, when partner modeling is removed, what serves as the input for the ego policy?
>
> When the partner modeling is removed, the ego policy only depends on the state and has the form $\pi_e(a|s)$. When we use partner modeling, the ego policy also depends on the predicted partner action distribution $a_p$ and has the form $\pi_e(a|s,a_p)$.
>
> > Q5: Minor comments: a) Figure 6 is difficult to comprehend. The importance of "environment steps" is not clear in the figure. I think it would be helpful to use the "accuracy of action prediction" as the x-axis, .......
>
> The environment step is the number of interactions used for training checkpoints. It is useful to plot the "average reward per episode" and the "action prediction" metrics over time to present how the agents improve their coordination and partner modeling skills. We have also illustrated the correlation between these two metrics in Figure 1. (a) of the rebuttal material, where the x-axis is the "action prediction" accuracy.

---

> ### Author Response · Authors · 2023-08-20
> **Providing additional experiment results on the Google Football game**
>
> Dear Reviewer dkSb,
>
> We really thank you for your valuable comments on improving our work. Here, we would like to add more experiment results and a more detailed analysis of a 3-player Google Football game.
>
> As per your suggestion, we have conducted additional experiments on the Google Football environment's "3 vs 1 with Keeper" layout, which is a three-player cooperative game with 19 actions in the discrete action space. The three players have shared rewards and they cooperate to gain high rewards. We train policies via self-play and our methods. To test the zero-shot coordination ability of these models, we follow the experimental setting of Other-Play [1]. In this setting, **the policies cooperate with unseen teammate policies that were trained with the same algorithms but different random seeds, without any prior interaction.** This ensures that the policies have not encountered their teammates during training. In the table below, we report the mean and standard error of the winning rates of scoring a goal.
>
> |                                       | Training performance | Test performance(Zero-shot coordination  across different random seeds) |
> | ------------------------------------- | -------------------- | ------------------------------------------------------------ |
> | Self-Play                             | 0.87(0.05)           | 0.03(0.01)                                                   |
> | E3T(Mixture Policy)                   | 0.79(0.04)           | 0.70(0.01)                                                   |
> | E3T(Partner Modeling)                  | 0.89(0.05)           | 0.24(0.05)                                                   |
> | E3T(Mixture Policy + Partner Modeling) | 0.87(0.02)           | 0.84(0.02)                                                   |
>
> As shown in the table above,  policies trained by the self-play fail to cooperate with policies from different training seeds, because self-play policies may fall into some specific coordination conventions during training. Our proposed mixture policy can increase the behavioral diversity of teammates, allowing the ego policy to encounter different coordination patterns during training. Consequently, our method with the mixture policy can train a policy that can adapt well to policies independently trained from different training seeds. Moreover, the partner modeling module can further enhance the zero-shot coordination performance by enabling the ego policy to respond to the predicted teammates' action distributions. We note that we set the epsilon as 0.1 for this environment (a large epsilon, i.e., more randomness, would result in policies failing to obtain rewards on this task) and the partner modeling module uses the historical observations of the ego policy to predict the action distributions of all 3 players.
>
> In summary, our method combining the mixture partner policy and partner(teammate) modeling module can also improve the zero-shot coordination ability in the complex football environment, with 3 cooperative players and a large action space with 19 discrete actions in it.
>
>  [1] Hu, H., Lerer, A., Peysakhovich, A., and Foerster, J. “other-play” for zero-shot coordination. In International Conference on Machine Learning, pp. 4399–4410. PMLR, 2020.
>
> We hope that our responses have solved your concerns and respectfully hope you can consider the final decision accordingly. As the author-reviewer discussion period is coming to an end, we are respectfully looking forward to discussing with you. Please let us know if you have any further questions or concerns and we are very happy to address them.

---

> > ### Comment · Reviewer_dkSb · 2023-08-22
> > **Official Comment by Reviewer dkSb**
> >
> > I would thank the authors for their rebuttal and additional experiments. The rebuttal effectively tackled the majority of my concerns and inquiries. As a result, I would like to raise my score to acceptance.

---

### Author Rebuttal · Authors · 2023-08-10

We thank the time and effort reviewers have invested in reviewing our paper. We have provided detailed explanations and clarifications to resolve your concerns regarding experiments and insights. If you have further concerns, please feel free to respond to us and we would like to discuss them with you.

We have conducted further experiments and ablations to address your comments and concerns. The results are presented in the rebuttal PDF file.

**R1: Common response to Reviewers dkSb and kGQD. Additional experiments on Google Football**
To further verify the effectiveness of our method, we conduct additional experiments on the Google Football environment's "3 vs 1 with Keeper" layout, which is a three-player cooperative game with 19 actions in the discrete action space. Following the Other-Play [1], we use the coordination performance among independent training runs ("random seeds") from the same training method to evaluate the zero-shot coordination ability. We report the mean and standard error of the winning rates of scoring a goal in the table below.

|                     | Training performance | Test performance Zero-shot coordination  across different random seeds |
| ------------------- | -------------------- | ------------------------------------------------------------ |
| Self-Play           | 0.87(0.05)           | 0.03(0.01)                                                   |
| E3T(mixture policy) | 0.79(0.04)           | 0.70(0.01)                                                   |

These results show that the policy trained by self-play can coordinate well with itself but it fails to cooperate with policies trained with different random seeds (with different behavior preferences). However, the policy trained by our method (with a mixture policy) can cooperate well with itself and with policies trained under other random seeds. These results indicate that increasing partner diversity during training can significantly improve the generalization ability of trained ego policy.

[1] Hu, H., Lerer, A., Peysakhovich, A., and Foerster, J. “other- play” for zero-shot coordination. In International Conference on Machine Learning, pp. 4399–4410. PMLR, 2020.

---

### Comment · Area_Chair_j7dw · 2023-08-11
**Discussion with authors**

Dear Reviewers,

Please check the rebuttal and start a discussion with authors if you need any additional information to make your final decision. The discussions should be completed by Aug 16.

Thanks,

AC

---

### Decision · Program_Chairs · 2023-09-21

**Decision:**

Accept (poster)

**Comment:**

The paper proposes an approach for zero-shot coordination with humans, where an agent adapts to different partner behaviors in a collaborative task. The reviewers initially had several concerns including:

(1) Limited set of experiments (e.g., using only 5 layouts)

(2) Limited novelty

(3) Sensitivity of the performance to the parameters

(4) Strong assumptions such as having access to the partner’s state-action pairs

The rebuttal addressed the concerns very well leading to an increase of the rating by some of the reviewers. Despite the weaknesses (such as running experiments in simple environments), the AC believes the paper has merit and the insights are useful for human-AI coordination tasks. Hence, acceptance is recommended.